# LaRes: Evolutionary Reinforcement Learning with LLM-based Adaptive Reward Search

**Pengyi Li**
College of Intelligence and Computing
Tianjin University
lipengyi@tju.edu.cn

**Hongyao Tang**
College of Intelligence and Computing
Tianjin University
tanghongyao@tju.edu.cn

**Jinbin Qiao**
College of Intelligence and Computing
Tianjin University
jinbin@tju.edu.cn

**Yan Zheng†**
College of Intelligence and Computing
Tianjin University
yanzheng@tju.edu.cn

**Jianye Hao†**
College of Intelligence and Computing
Tianjin University
jianye.hao@tju.edu.cn

## Abstract

The integration of evolutionary algorithms (EAs) with reinforcement learning (RL) has shown superior performance compared to standalone methods. However, previous research focuses on exploration in policy parameter space, while overlooking the reward function search. To bridge this gap, we propose **LaRes**, a novel hybrid framework that achieves efficient policy learning through reward function search. LaRes leverages large language models (LLMs) to generate the reward function population, guiding RL in policy learning. The reward functions are evaluated by the policy performance and improved through LLMs. To improve sample efficiency, LaRes employs a shared experience buffer that collects experiences from all policies, with each experience containing rewards from all reward functions. Upon reward function updates, the rewards of experiences are relabeled, enabling efficient use of historical data. Furthermore, we introduce a Thompson sampling-based selection mechanism that enables more efficient elite interaction. To prevent policy collapse when improving reward functions, we propose the reward scaling and parameter constraint mechanisms to efficiently coordinate reward search with policy learning. Across both initialized and non-initialized settings, LaRes consistently achieves state-of-the-art performance, outperforming strong baselines in both sample efficiency and final performance. The code is available at https://github.com/yeshenpy/LaRes.

## 1 Introduction

Reinforcement learning (RL) [1] is a class of learning methods that excels at handling sequential decision-making problems [2]. Through trial and error and gradient-based optimization, RL approximates value functions and provides policy gradients for policy learning [3]. It has been applied in various fields, including robotic control [4], game AI [5], and recommender systems [6]. In contrast,

---

†Corresponding authors: Yan Zheng and Jianye Hao

39th Conference on Neural Information Processing Systems (NeurIPS 2025).

Evolutionary Algorithms (EAs) [7–9] are heuristic optimization methods inspired by Darwinian principles, typically employing gradient-free approaches to solve problems and have shown remarkable performance in fields like circuit design [10] and scheduling optimization tasks [11]. Previous studies have revealed complementary characteristics between these two approaches [12, 13]. RL excels at utilizing fine-grained information, such as states and actions, offering high sample efficiency and strong local optimization capabilities. However, it faces exploration challenge and is prone to suboptimal solutions [14]. In contrast, EAs are strong in global optimization but suffer from weak local optimization and sample inefficiency [15, 16]. Given these complementary strengths, many works have explored the integration of EAs and RL for policy learning, demonstrating superior performance compared to each approach individually [17–20].

The ultimate goal of task solving is to learn an efficient policy, which depends on two key factors: the algorithm's search capability in the policy parameter space, and the quality of the task's reward function, which directly impacts policy learning performance and efficiency [1]. Previous works primarily focus on integrating EAs and RL to improve the policy search capabilities [13], with little attention given to the reward functions. Early works aim to improve reward functions through heuristic operators, but these methods struggle to scale to complex tasks [21]. With the advancement of large language models (LLMs) that demonstrate strong coding capabilities and valuable domain knowledge, the generation of complex reward functions using LLMs has been explored preliminarily. For example, Text2Reward [22] uses LLMs to construct reward functions based on structured environment representation. Eureka [23] constructs a reward function population and improves the reward function based on evolutionary principles. SA [24] integrates CoT and hyperparameter optimization. R* [25] further improves Eureka's performance through structural evolution and parameter optimization. However, these works overlook sample efficiency, a key evaluation metric in RL, leading to excessive environment interactions. In contrast, LaRes focuses on sample-efficient policy learning from both the policy and sample perspectives.

To solve these problems, we propose an **LLM**-based **a**daptive **Re**ward **s**earch hybrid framework (**LaRes**) for efficient policy learning. In LaRes, we use LLMs to generate a population of candidate reward functions, while RL learns corresponding policies. The LLM then iteratively refines the reward population based on the performance feedback from these learned policies. If a human-designed reward function is available, it can be included in the context to reduce the difficulty of reward search. To improve sample efficiency, we maintain a shared replay buffer that stores experiences from all policies. **The key difference from previous works is that each experience includes multiple rewards from the reward function population, rather than a single reward.** RL then optimizes multiple policies based on the corresponding rewards. Besides, when the reward functions are improved, we (1) relabel the corresponding rewards in the replay buffer to enable historical data reuse, and (2) allow RL individuals to inherit from the best individual, avoiding retraining from scratch. However, we find that reward function changes may lead to policy collapse. To stabilize learning, we propose two mechanisms: reward scaling, which aligns the scale of the new reward function with the elite one, and parameter constraint loss, which minimizes the distance between the policy and critic and their elite counterparts in parameter space. Moreover, different reward functions may result in significant performance variations among policies, the experiences from inferior policies can contaminate the replay buffer, leading to suboptimal policies. Thus we propose Thompson sampling [26]-based interaction mechanism, which prioritizes more frequent interactions for superior policies. In experiments on 16 robotic manipulation and 4 MinAtar tasks with human-designed reward initialization, LaRes outperforms strong RL, ERL, and reward design baselines in both sample efficiency and final performance. When trained without initialization, LaRes continues to achieve state-of-the-art results in manipulation and locomotion tasks.

We summarize our contributions as follows: (1) We propose a novel hybrid framework that focuses on improving sample efficiency from both the sample and policy perspectives. (2) From the sample perspective, we design a shared replay buffer with a reward relabeling mechanism to fully utilize historical data. (3) From the policy perspective, we propose reward scaling and parameter constraint mechanisms to ensure training stability, and a Thompson sampling–based interaction mechanism to balance exploration and exploitation. (4) Empirical results show that LaRes consistently outperforms strong baselines across a wide range of tasks and experimental settings.

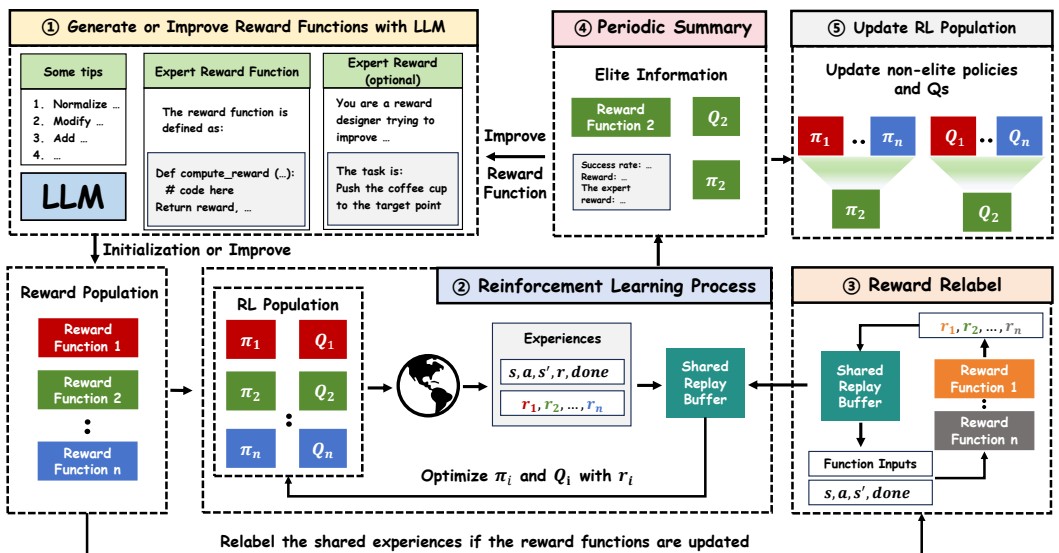

Figure 1: The optimization flow of LaRes. In the first iteration, ① LaRes generates the reward function population using the LLM, followed by ② RL training. After a certain number of training steps, ④ results are summarized, and then ⑤ the non-elite RL individuals are replaced by the elite ones. Next, ① the reward function population is enhanced based on the best reward function, followed by ③ the reward relabel phase. The subsequent iterations follow the sequence: ②, ④, ⑤, ①, and ③.

## 2  Background

**Reinforcement Learning**  Consider a Markov decision process (MDP) [1], defined by a tuple $\langle \mathcal{S}, \mathcal{A}, \mathcal{P}, \mathcal{R}, \gamma, T \rangle$. At each step $t$, the agent uses a policy $\pi$ to select an action $a_t \sim \pi(\cdot|s_t) \in \mathcal{A}$ according to the state $s_t \in \mathcal{S}$ and the environment transits to the next state $s_{t+1}$ according to transition function $\mathcal{P}(s_t, a_t)$ and the agent receives a reward $r_t = \mathcal{R}(s_t, a_t)$. The return is defined as the discounted cumulative reward, denoted by $R_t = \sum_{i=t}^{T} \gamma^{i-t} r_i$ where $\gamma \in [0, 1)$ is the discount factor and $T$ is the maximum episode horizon. The goal of RL is to learn an optimal policy $\pi^*$ that maximizes the expected return. SAC [27] is one representative RL algorithm that maximizes expected reward while maintaining high entropy for effective exploration. SAC maintains a policy and double Q-networks, with the policy optimized as follows:

$$\mathcal{L}_\pi = \mathbb{E}_{\mathcal{D},\pi}\big[\alpha \log \pi(a|s) - \min_i Q_{\phi_i}(s,a)\big], \quad \mathcal{L}_Q(\phi_i) = \mathbb{E}_{\mathcal{D},\pi}\big[Q_{\phi_i}(s,a) - (r + \gamma V'(s'))\big]^2,$$
(1)

where $V'(s') = \mathbb{E}_\pi\big[\min_j Q_{\hat{\phi}_j}(s',a') - \alpha \log \pi(a'|s')\big]$ and $\alpha$ is the temperature parameter.

**Evolutionary Algorithm**  Evolutionary Algorithms (EAs) [7, 28, 9] are a class of black-box optimization methods. EAs typically need to maintain a population of individuals. In previous hybrid works, the population individuals are often represented in various forms [13], e.g., policy networks, value function networks, and the evolution of the population is achieved through crossover and mutation operations on the parameters. In this paper, the population maintained by the EAs exists in the form of reward function code, defined as follows: $\mathbb{P} = \{F_1, F_2, ..., F_n\}$. The reward function fitness $f(F_i)$ is evaluated based on the performance of the policy guided by $F_i$. We improve the reward function through LLMs.

## 3  Related Work

The integration of EAs with RL has demonstrated strong capabilities across various tasks [13, 16]. Some works incorporate RL into EAs to improve population initialization [29], evolutionary operators [30], and other processes [31]. Other works integrate EAs into RL for hyperparameter tuning [32], action selection [33], and exploration [34]. Another emerging active direction is to fuse

the strengths of both approaches. For example, methods such as ERL [12], PDERL [18], CEM-RL [17], ERL-Re$^2$ [19], EvoRainbow [20] leverage the complementary strengths of EAs and RL, improving both sample efficiency and exploration in policy search. Other works, like VFS [35] and VEB-RL [36], focus on optimizing value functions. In addition, some works have extended ERL concepts to areas like multi-agent systems [37, 38] and game testing [39]. In contrast, LaRes focuses on improving the reward function to enhance policy learning efficiency in complex tasks.

Early works on reward design primarily focus on inverse RL [40–43], where rewards are constructed based on expert demonstrations. Subsequently, the emergence of RL from human feedback (RLHF) enables learning reward models directly from human feedback [44–48]. In addition, some works improve reward quality through reward shaping [49–53]. With broad domain knowledge and strong coding capabilities [54–59], LLMs have been successfully applied across various fields [60–65]. Among them, recent works explore leveraging LLMs to generate reward functions. For example, L2R [66] employs LLMs to write reward code based on predefined APIs. Eureka [23] follows an evolutionary approach, maintaining a population of reward functions to guide policy learning and iteratively refining them with an LLM. DrEureka [67] uses LLMs to write reward functions and configure domain randomization parameters to achieve sim-to-real transfer. R* [25] decomposes reward design into structural evolution and parameter optimization. However, these works overlook sample efficiency, a key evaluation metric in RL. In contrast, LaRes focuses on improving sample efficiency from both the sample and policy perspectives.

## 4 LLM-based Adaptive Reward Search

This section provides an overview of LaRes and elaborates its key mechanisms from three perspectives: high-level reward evolution, low-level policy learning, and inter-level coordination.

### 4.1 LaRes Optimization Flow

Unlike previous ERL methods that use EAs and RL to co-optimize the policy parameters, LaRes decomposes the task-solving process into reward function search and policy learning. The overall framework is shown in Figure 1. LaRes employs LLM to construct and evolve the reward function population. We then employ RL to learn the corresponding policies and evaluate the reward function fitness based on the policy performance. Below, we briefly summarize the algorithmic process, from high-level reward evolution, lower-level policy learning, and coordination between the two levels.

**Population initialization.** In the initial iteration, we provide human-designed reward functions (optional), task descriptions, and environment information as the context to the LLM. Using tailored prompts, the LLM generates $n$ reward functions to construct the initial reward function population. For each reward function, an RL agent is initialized for environment interaction and policy learning.

**Population Evaluation and Evolution.** With $T$-step learning, we select the best reward function together with its corresponding policy and Q-function based on policy performance, e.g. success rate. Subsequently, we provide the best reward function and learning process information (e.g., success rates, cumulative rewards) as feedback to the LLM for reflection. The LLM then generates new improved reward functions that replace the non-elite ones in the population.

The above process introduces the high-level reward search, focusing on reward function generation, evaluation and evolution. Below, we introduce the low-level policy learning.

**Policy Learning.** We employ the off-policy RL algorithms (e.g., SAC [27]) to learn the policies guided by different reward functions. To improve sample efficiency and mitigate exploration challenges in RL, experiences from all policies are stored in a shared replay buffer. To reduce replay buffer contamination from inferior policy experiences, we propose the selection-based interaction using Thompson sampling [26]. Further details are provided in Subsection 4.3.

The coordination between the two levels is crucial, with a particular focus on how policy learning adapts to the periodic evolution of the reward function.

**Continual Learning.** After the reward function is improved, continuing learning with the previous policy and Q-functions may lead to suboptimal results, while learning a new RL policy from scratch incurs a significant sample cost. Thus we propose a continual learning approach, where the best policy and its corresponding Q-functions are used to initialize those of the non-elite agents. However,

we find that the elite policy is prone to performance collapse when guided by new reward functions. To solve the problem, we propose the reward scaling and parameter constraint mechanisms, which will be discussed in Subsection 4.3.

**Historical Data Reuse.** Off-policy RL algorithms like SAC can reuse historical data through a replay buffer. However, when the reward function changes, directly training on previous data may lead to policy collapse due to reward inconsistency problem, while discarding historical data would result in significant sample waste. To efficiently reuse historical data, we relabel the shared replay buffer based on the new reward function population.

Next, we present a detailed introduction to the key components of LaRes.

## 4.2 High-level Reward Evolution

Inspired by previous works [22, 23], we provide the raw environment variables as context to the LLM, along with tailored prompts that guide it in generating and refining reward functions. The LLM then iteratively generates $n$ reward functions in code format, which are subsequently optimized from three key perspectives: 1) **Components Optimization**. Adding new reward components for guidance or removing ineffective reward components. 2) **Weight Optimization**. Adjusting the weights between different reward components. 3) **Calculation Optimization**. Modifying the calculation method of reward components, such as using alternative activation functions.

The LLM must consider all three aspects simultaneously. If optimization is not performed, a brief explanation should be provided. In the improvement phase, the best reward function is selected according to the policy performance. We then provide the policy performance, episodic reward, and other related statistics to the LLM, which reflects on this feedback and generates new reward functions to replace the non-elite ones in the population. Through the above process, we can achieve the iterative improvement of the reward function population. To ensure the stability of policy learning, the elite reward functions will not be replaced and continue to guide their corresponding RL agents. Additionally, under settings with human-designed reward initialization, we always maintain the human-designed reward function, resulting in a total of $n + 1$ reward functions.

## 4.3 Low-level Policy Learning

Given the presence of multiple reward functions, we employ the parallel training approach, learning an RL agent for each reward function. To improve sample efficiency, we maintain a shared replay buffer that stores the experiences from all policies. The key difference is that, the conventional replay buffer stores experiences in the format $\{s, a, s', r, \text{done}\}$, due to the presence of the reward function population, each experience needs to be relabeled by the reward function population, resulting in the format $\{s, a, s', r, r_1, \ldots, r_n, \text{done}, \text{info}\}$, $r$ represents the original human-designed reward, which exists only when a human-designed reward function is provided, $\{r_1, \ldots, r_n\}$ represent the rewards from the population, and info denotes the variable inputs to the reward function for reward calculation. Each RL agent is trained using its corresponding rewards.

In addition, due to the significant performance differences in policies guided by different reward functions, allocating the same number of interaction steps to each agent would lead to resource waste. To improve the experience quality in the replay buffer while mitigating the impact of poor experiences, we propose a Thompson sampling-based interaction mechanism. Thompson sampling [26] is a Bayesian inference-based sampling method that estimates the distribution of the expected reward for each action based on historical data. Actions are selected by sampling from these distributions, and after performing the action, the reward distribution is updated using the reward feedback. In LaRes, we define $n$ actions, where the $i$-th action corresponds to selecting policy $i$ for interaction. We update the Beta distribution parameters using feedback from task success and failure. Specifically, the reward $\psi_i$ of each action follows Beta distribution $\psi_i \sim \text{Beta}(\alpha_i, \beta_i)$, and the action with the highest sampled reward is selected for interaction. For action $i$, $\beta_i$ and $\alpha_i$ are updated as follows:

$$\alpha_i' = \alpha_i + n_{s,i}, \beta_i' = \beta_i + n_{f,i}, \tag{2}$$

where $n_{s,i}$ and $n_{f,i}$ represent the number of successes and failures of policy $i$, respectively. Under this mechanism, superior policies have more interaction resources, while inferior policies have fewer opportunities for interaction. Additionally, after the reward function improves, we reset the Thompson sampler to reallocate resources. Note that following previous ERL works, the RL agent guided by the human-designed reward function does not participate in sampling and interacts in each iteration.

The iterative improvement of the reward function population can introduce several problems for low-level policy learning. In this subsection, we provide a detailed explanation of how to address these challenges. When the reward function changes, we adopt a continual learning approach to directly configure the best policy and Q functions for the new reward function. This approach avoids the high sample cost of training from scratch and mitigates the performance degradation caused by continuing to train an inferior policy. However, we find that when the reward function changes, policy learning is prone to collapse. This is primarily caused by two factors: reward scale difference and reward design difference.

To address the scale difference, we introduce a reward scaling mechanism. Specifically, we calculate the mean $\mu_{\text{elite}}$ and variance $\sigma_{\text{elite}}$ of the elite reward function based on the replay buffer as a surrogate for its true mean and variance. Meanwhile, we calculate the mean $\mu_{\text{new}}$ and variance $\sigma_{\text{new}}$ of the newly generated reward function. We align the means and variances of the two reward functions using the following formula:

$$r_{\text{scaled}} = \frac{\sigma_{\text{elite}}}{\sigma_{\text{new}}} \left( r_{\text{new}} - \mu_{\text{new}} \right) + \mu_{\text{elite}}. \tag{3}$$

By scaling both the replay buffer rewards and the new interaction rewards, we can efficiently address the reward scale difference problem. However, distributional differences still exist. In previous ERL works, evolution typically occurs at the parameter level, under the assumption that better individuals are located near the parameters of the optimal individual. Inspired by this, we propose parameter constraint mechanism that constrains the parameter changes of the policy and value functions to further mitigate the collapse problem. Specifically, we introduce the following loss:

$$\mathcal{L}_{\pi_i} = \|\theta_i - \theta_{\text{elite}}\|_2^2, \mathcal{L}_{Q_i} = \|\phi_{0,i} - \phi_{0,\text{elite}}\|_2^2 + \|\phi_{1,i} - \phi_{1,\text{elite}}\|_2^2, \tag{4}$$

where $\theta_{\text{elite}}, \phi_{0,\text{elite}}$ and $\phi_{1,\text{elite}}$ are the policy parameters and value function parameters of the elite agent. By adding the above constraint loss during both policy and value function updates, we prevent the policy from drifting away from the elite parameters, thereby improving training stability.

## 4.4 LaRes Algorithm

LaRes is a flexible framework that can be combined with any off-policy RL method. We provide the pseudocode in Algorithm 1. Specifically, we first use the LLM to generate a reward function population based on the human-designed reward function $F_H$ (in human reward function initialization setting), environment variables $V$, and other information (line 3). In each iteration, we first initialize the Thompson sampler $T_S$ (line 5). Next, we enter the training phase, which begins with policy interactions (line 6-11), including both the $T_S$ sampled policy and the policy guided by the human-designed reward (line 7). For each experience, rewards are calculated using $\mathbb{P}_{\text{Reward}}$ (line 8), and all experiences are added to the shared replay buffer $D$ (line 9). Subsequently, parallel RL training is conducted (line 10). If the current iteration is not the initial iteration, a parameter constraint loss is added to the non-elite agent training process to ensure learning stability. After $T$ environment steps of

---

**Algorithm 1** LaRes Framework

1: **Require**: Task description $L$, environment variables $V$, reward function prompt $P$, coding LLM $LLM$, Human-designed reward function $F_H$ (Optional)
2: **Initialize** Shared Replay Buffer $D$, Thompson sampler $T_S$, an RL population $\mathbb{P}_{\text{RL}} = \{\pi_1, Q_1, \cdot, \pi_n, Q_n\}$
3: Initialize reward function population:
$\mathbb{P}_{\text{Reward}} = LLM(L, V, P, F_H(optional))$
4: **for** $N$ Iterations **do**
5:    Reset the Thompson sampler $T_S$
6:    **for** $T$ environment steps **do**
7:       Policy interaction based on sampler $T_S$
8:       Add all rewards for each experience using $\mathbb{P}_{\text{Reward}}$
9:       Add all experiences to $D$
10:      RL parallel training, add parameter constraint loss to non-elite agents
11:   **end for**
12:   Select the best Reward Function $F_{\text{best}}$, $\pi_{\text{best}}$ and $Q_{\text{best}}$
13:   Improve reward population with LLM reflection
$\mathbb{P}_{\text{Reward}} = \text{Reflection}(L, V, P, F_H(optional), F_{\text{best}})$
14:   Relabel replay buffer $D$ using the new $\mathbb{P}_{\text{Reward}}$
15:   Initialize the non-elite RL agents with the best $\pi_{\text{best}}$ and $Q_{\text{best}}$
16:   Reward rescaling for new reward functions
17: **end for**

---

training, we evaluate the performance of the learned RL policies and select the best reward function $F_{\text{best}}$, together with its corresponding policy $\pi_{\text{best}}$ and value function $Q_{\text{best}}$(line 12). Using the LLM

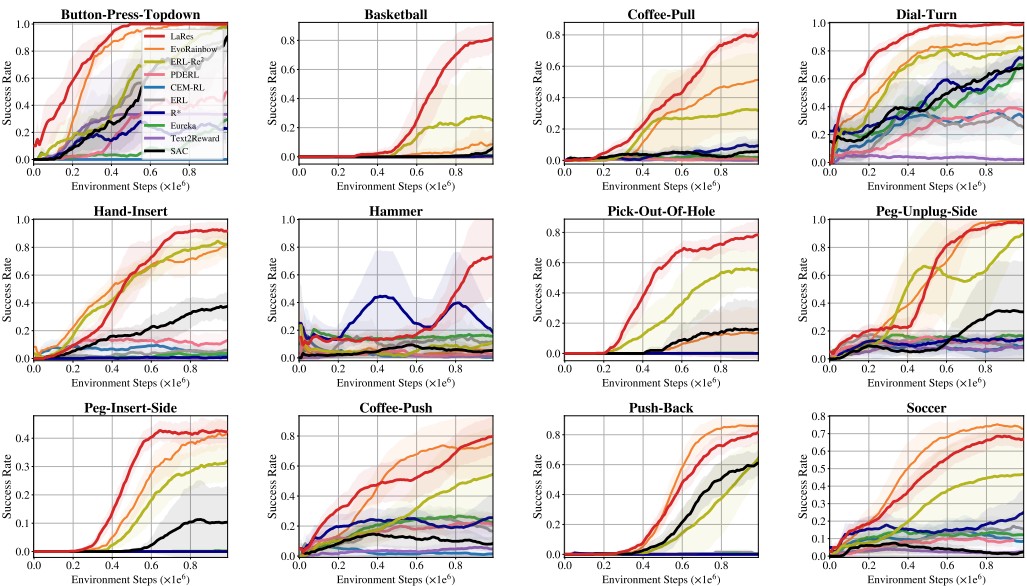

Figure 2: Performance comparison on 12 robot manipulation tasks from MetaWorld.

reflection mechanism, we improve the reward function population by replacing non-elite individuals (line 13). The experiences in $D$ are then relabeled based on the new reward function (line 14), and the parameters of the non-elite individuals are initialized using $\pi_{\text{best}}$ and $Q_{\text{best}}$ (line 15). To ensure reward scale consistency, we apply a reward rescaling mechanism to adjust the reward function (line 16). The process then proceeds to the next iteration. Through the above process, LaRes achieves efficient policy learning through reward search.

## 5 Experiments

We first conduct experiments on various tasks to compare LaRes with other strong baselines. To gain deeper insights into LaRes, we then perform detailed analyses. Furthermore, an ablation study is conducted to verify the effectiveness of each component.

### 5.1 Experimental Setups

We evaluate LaRes on a wide range of benchmarks, including manipulation tasks from the MetaWorld and ManiSkill3 suites [68, 69], MinAtar tasks with image inputs [70], and locomotion tasks from MuJoCo [71]. Under the setting with human-designed reward initialization, we evaluate LaRes on 20 tasks, including 12 MetaWorld tasks, 4 ManiSkill tasks, and 4 MinAtar tasks. In manipulation tasks, we implement LaRes based on SAC. We compare LaRes with the following baselines: 1) **RL baselines**, i.e., SAC [27]; 2) **ERL-related baselines**, including ERL [12], PDERL [18], CEM-RL [17], ERL-Re$^2$ [19], EvoRainbow [20]. 3) **Reward-search baselines**, i.e., SAC-based Eureka [23] and Text2Reward [22], R* [25]. In MinAtar tasks, we implement LaRes based on DQN and compare it with DQN. Under the no-initialization setting, we compare LaRes with other reward design methods on 5 MetaWorld tasks and 2 locomotion tasks (Ant & Humanoid).

We use the official codes or implement the methods on new benchmarks following the original papers. For a fair comparison, we fine-tune them in each task to provide the best performance. We use GPT-4o-mini as the LLM backbone under the human-designed reward initialization setting, and GPT-4o under the no-initialization setting. All algorithms are trained with 1 million environment steps on MetaWorld and locomotion tasks, 2 million environment steps on ManiSkill3 and MinAtar. All statistics are obtained from 5 independent runs, consistent with previous literature. We report the average with 95% confidence intervals. For LaRes, we set the population size to 5. We perform 5 iterations of the reward population evolution. All implementation details are provided in Appendix B.

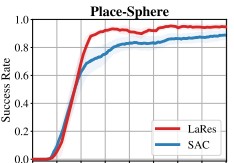
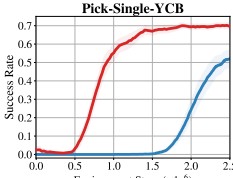
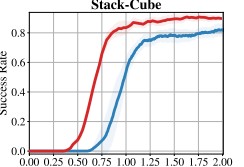
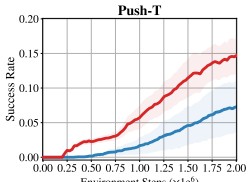

Figure 3: Performance comparison on 4 robot manipulation tasks from ManiSkill3.

| Method | Window-Close | Window-Open | Drawer-Open | Button-Press | Door-Close |
|---|---|---|---|---|---|
| LLM zero-shot | 51% ǀ 388796 | 6% ǀ 419084 | 8% ǀ 762346 | 65% ǀ 560428 | 15% ǀ 156590 |
| Eureka | 50% ǀ 303786 | 55% ǀ 657098 | 15% ǀ 731876 | 61% ǀ 924163 | 98% ǀ 283206 |
| LaRes | **100% ǀ 164850** | **100% ǀ 358403** | **100% ǀ 164850** | **100% ǀ 112800** | **100% ǀ 65626** |

Table 2: Comparison under the no-initialization setting (*Success Rate ǀ Samples Needed*).

## 5.2 Performance Evaluation with Human Reward Function Initialization

We begin by evaluating LaRes and other baselines across 12 different manipulation tasks in the MetaWorld benchmark. The RL and ERL baselines learn policies guided by human-designed rewards. The experimental results are shown in Figure 2. We observe that LaRes significantly outperforms SAC (guided by human-designed rewards) across all tasks, which demonstrates that LaRes effectively discovers better reward functions and fully leverages the learned multiple policies to achieve superior performance. Furthermore, LaRes outperforms other ERL methods in both sample efficiency and final performance across most tasks, especially the harder ones. Notably, unlike other ERL methods, LaRes does not combine EAs and RL for co-optimizing policy and value function parameters. Instead, it focuses on reward function search and continuous policy learning. The results highlight both the importance of reward function design and the effectiveness of LaRes. Finally, we observe that LaRes significantly outperforms other reward design methods. Compared to Eureka and R*, LaRes makes more efficient use of the generated data and ensures stable policy learning, while Text2Reward lacks the capacity for continuous improvement, making the policy more prone to suboptimality or collapse.

We further evaluate LaRes and SAC on four tasks from ManiSkill3. Unlike MetaWorld, these tasks leverage GPU parallel sampling, enabling higher sampling efficiency. The results presented in Figure 3 demonstrate that LaRes can also significantly improve SAC. This demonstrates that LaRes consistently improves performance across different benchmarks, further validating the effectiveness of LaRes.

**Can LaRes improve other off-policy algorithms and be applied to discrete action spaces?** To verify this, we integrate LaRes with DQN and evaluate on 4 discrete-control tasks from

| Method | Breakout | Asterix | Freeway | SpaceInvaders |
|---|---|---|---|---|
| DQN | 13.52 ǀ 21.86 | 2.96 ǀ 11.55 | 38.89 ǀ 53.93 | 23.18 ǀ 65.38 |
| LaRes | **21.87 ǀ 31.28** | **14.13 ǀ 28.18** | **50.87 ǀ 57.24** | **42.27 ǀ 84.72** |

Table 1: The scores at 0.5 & 2 million env steps on MinAtar tasks.

MinAtar, and report the average scores at 0.5 and 2 million environment steps in Table 1. We observe that LaRes significantly improves the performance of DQN trained with human-designed rewards. This result further validates the effectiveness and generality of LaRes.

## 5.3 Performance Evaluation without Human Reward Functions

In the previous subsection, we mainly explore the results under the setting where a human-designed reward function is provided as initialization. A natural question arises: **Can LaRes still outperform other reward design methods when no human-designed reward function is available?** To answer this question, we first evaluate LaRes on 5 tasks from MetaWorld, which are similar to those used in Text2Reward. Compared to the tasks in the previous subsection, these tasks are relatively easier, allowing policies to achieve a 100% success rate. The experimental results are presented in Table 2. We observe that LaRes outperforms other methods in both sample efficiency and best performance, which indicates that LaRes is also efficient in its approach to learning from scratch.

Beyond the above manipulation tasks, we further evaluate LaRes on two challenging locomotion tasks, Humanoid and Ant. As shown in Table 3, LaRes again surpasses other reward design methods, demonstrating its effectiveness across different tasks.

| Speed 1M (m/s) | Eureka | ROSKA [72] | LaRes |
|---|---|---|---|
| Ant | 1.49 | 2.77 | **4.44** |
| Humanoid | 2.02 | 2.95 | **3.42** |

Table 3: Performance on locomotion tasks.

## 5.4 Analysis & Ablation

In this section, we answer the following questions through experiments: **Q1**. Does LaRes effectively improve the reward function? **Q2**. Does the Thompson sampler in LaRes efficiently and dynamically adjust the policy interaction? **Q3**. Can the reward scaling and parameter constraint mechanisms effectively prevent policy collapse?

To answer Q1, we present performance comparison between the policy guided by the human-designed reward and the best one guided by the generated reward function in LaRes. The results are shown in Figure 4. We observe that the generated reward function significantly outperforms the human-designed reward function both in terms of sample efficiency and final

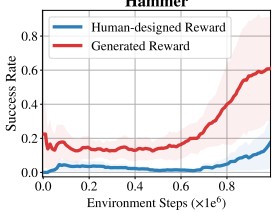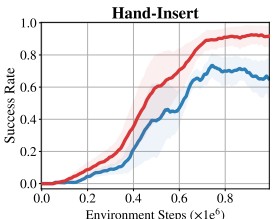

Figure 4: Performance comparison of the policy success rate guided by human-designed rewards and generated rewards.

performance. This indicates that LaRes can discover higher-quality reward function than the human-designed expert reward function. Additionally, we provide the LLM summary of the best reward function generated on the Hammer task. Due to space limitations, the complete reward functions are provided in Appendix C. We observe that the LLM is able to make reasonable improvements based on the three designed aspects, enabling more efficient exploration of the reward space.

---

**LLM Summary of Improved Reward Function**

1. **Reward Components**: The reward for lifting the hammer ('a') and for hitting the target nail ('b') was increased significantly to enhance the focus on these critical actions. The increased penalties when nearing the target position (e.g. adjusted bounds to '0.015') will encourage more precise interactions with the target area.
2. **Reward Weights**: The weight of the successful lift and placement were modified to provide enhanced emphasis on these actions, thereby creating a stronger incentive for the agent to accomplish these milestones, which seems crucial considering the policy's current low success rate.
3. **Reward Calculation**: The overall normalization and adjustment of reward components now includes changes in thresholds and the temperature parameter, which should provide more stability and control in training. Lowering the temperature value improves the agent's control over the learning process and minimizes excessive reward scaling.

---

To answer Q2, we first present detailed learning curves of policies guided by different functions on the Hammer task. We observe that the policies show significant differences, which aligns with the intuition that the modifications guided by LLMs do not always yield positive results. Then we visualize the selected probability of each policy using Thompson sampler. We can observe that $F_3$ is selected with a high probability in each iteration, while $F_5$, due to its inferior

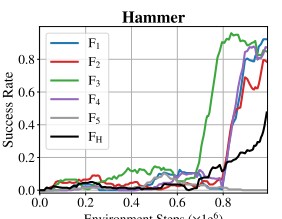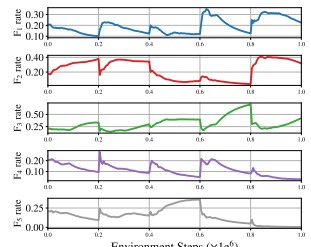

Figure 5: (Left) Policy learning performance guided by different reward functions, (Right) The variation curve of the probability of policy selection.

performance, has a low probability of being selected. Additionally, we conduct an ablation study on

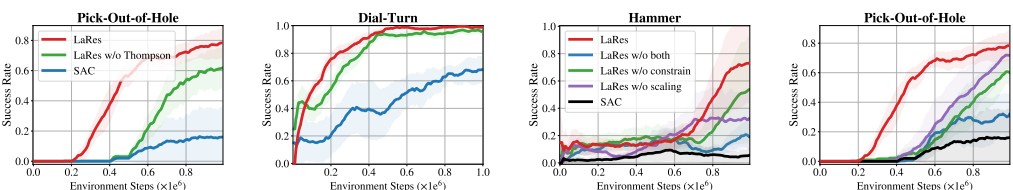

(a) Thompson-sampling interaction mechanism    (b) Reward scale & parameter constraint.

Figure 6: Ablation study on LaRes

the Thompson sampling mechanism. The results in Figure 6a show that removing Thompson sampling leads to a performance decline. This is consistent with the intuition that uniformly distributing interaction resources leads to poor performance.

To answer Q3, we conduct an ablation study on the reward scaling and parameter constraint mechanisms. The experimental results in Figure 6 demonstrate that LaRes w/o reward scaling and parameter constraints exhibits performance fluctuations during the learning process. However, adding reward scaling and parameter constraints leads to more stable and efficient policy learning, highlighting the effectiveness of these mechanisms. More experiments on hyperparameters and different LLM backbones are provided in Appendix D.

Finally, we present an overhead analysis. LaRes employs a parallel training architecture to simultaneously train multiple policies, effectively reducing the training time required for multi-policy learning. LaRes incurs an approximately 20% increase in time overhead, primarily due to inter-process communication. This overhead could be further reduced [73], such as adopting asynchronous communication. In addition, one limitation of LaRes is its increased computational resource requirements, which we aim to address in future work.

## 6   Conclusion

This paper introduces LaRes, a novel evolutionary reinforcement learning hybrid framework focused on reward function search. Driven by LLM, LaRes operates on three levels: reward function search at the high level, policy learning at the low level, and coordination between the two levels. Specifically, the high level employs the LLM to refine human-designed reward functions, generating improved reward functions. The low level employs Thompson sampling to adaptively select policies for interaction based on their performance and utilizes a shared replay buffer to enhance sample efficiency. The coordination between the two levels focuses on continuous policy learning. To achieve this, we propose a reward relabeling mechanism to efficiently reuse historical data, along with reward scaling and parameter constraint mechanisms to mitigate the policy-collapse problem. Across 16 robotic manipulation tasks, LaRes demonstrates significant improvements in both sample efficiency and final performance compared to other strong baselines.

## Acknowledgments

This work is supported by the National Natural Science Foundation of China (Grant Nos. 624B2101, 62422605, 92370132), National Key Research and Development Program of China (Grant No. 2024YFE0210900), and Xiaomi Young Talents Program of Xiaomi Foundation. We would like to thank all the anonymous reviewers for their valuable comments and constructive suggestions, which have greatly improved the quality of this paper.

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

# A  Limitations & Future Work

For limitations and future work, firstly our work is empirical proof of the effectiveness of the LaRes idea and we provide no theory on optimality, convergence, and complexity. Secondly, although LaRes adopts a parallel training architecture, it still incurs additional time overhead, primarily due to inter-process data communication. This overhead could be further optimized [73], for instance, by implementing an asynchronous communication mechanism. Furthermore, the computational cost of LaRes scales proportionally with the number of policies being trained. Thirdly, this work represents only an initial exploration of reward function search in the ERL framework. The proposed architecture can be further optimized, for example, by incorporating human-designed reward-guided policies into the Thompson sampling-based interactions or adopting more efficient sampling mechanism and continuous policy learning mechanism. Fourthly, LaRes does not explore more efficient generation mechanisms. For example, leveraging reasoning techniques within LLMs, such as Chain-of-Thought (CoT) [74] or Tree-of-Thought (ToT) [75] reasoning, could potentially enhance the quality of the generated reward functions. Finally, diversity is a key aspect of EA [76]. When constructing reward functions, we do not explicitly consider individual diversity and instead rely on the randomness of LLMs. However, this approach often fails to ensure sufficient differentiation among individuals.

Overall, LaRes represents a preliminary attempt at leveraging LLMs for reward function search within the ERL framework. However, it still has several limitations and potential areas for further improvement. We aim to address these challenges and enhance the method in future work.

# B  Method Implementation Details

The MetaWorld experiments are carried out on Intel(R) Xeon(R) CPU E5-2680 v4 @ 2.40GHz. ManiSkill3 leverages GPU acceleration; therefore, we conduct experiments on NVIDIA GTX 2080 Ti GPU with Intel(R) Xeon(R) CPU E5-2680 v4 @ 2.40GHz.

## B.1  Implementation of Baselines

For all ERL baseline algorithms, we use the official implementation, including ERL[1], PDERL[2], CEM-RL[3], ERL-Re2[4], EvoRainbow[5]. The reward-search baselines primarily include Text2Reward, Eureka and R*. For fair comparisons, all these methods are implemented based on the SAC algorithm with consistent hyperparameter settings. In addition, to handle discrete action spaces, we implement a DQN-based version of LaRes. For the implementation of Text2Reward, we employ the LLM to generate improved reward functions based on human-designed reward functions and interface definitions. Eureka can be regarded as a variant of LaRes without continual learning or data sharing. To achieve this, we modify the LaRes by removing unrelated components and ensuring the same number of evolutionary iterations and population size. R* further builds upon Eureka by introducing two mechanisms: parameter optimization and structure evolution.

## B.2  Implementation of LaRes

The implementation of LaRes has three versions: MetaWorld, ManiSkill3, and MinAtar. We build LaRes for MetaWorld based on the SAC implementation from EvoRainbow, for ManiSkill3 using the official SAC implementation provided by ManiSkill3, and for MinAtar using the official DQN implementation provided by VEB-RL[6].

For the training framework, we construct a parallel framework using Python's multiprocessing library. This framework consists of a central server and multiple workers. The server is primarily responsible for policy interactions, data distribution, parameter distribution, data labeling, and relabeling. The number of workers corresponds to the number of reward functions, and each worker is tasked with

---

[1] https://github.com/ShawK91/Evolutionary-Reinforcement-Learning

[2] https://github.com/crisbodnar/pderl

[3] https://github.com/apourchot/CEM-RL

[4] https://github.com/yeshenpy/ERL-Re2

[5] https://github.com/yeshenpy/EvoRainbow

[6] https://github.com/yeshenpy/VEB-RL

training a specific policy and sending the trained parameters back to the server for interactions and computations.

In all experiments, we maintain a reward function population of size 5, along with the original human-designed reward function, resulting in a total of 6 reward functions. The improvement of the reward function population involved a total of 5 evolutions (including the generation of the initial population). The elite size is set to 3 for all tasks. Thompson sampling parameters $\alpha$ and $\beta$ are set to 1 by default.

For all tasks in MetaWorld, we trained for 1 million environment steps, while for all tasks in ManiSkill3, we trained for 2 million environment steps. Consequently, for tasks in MetaWorld, population evolution is performed every 200,000 environment steps, whereas for tasks in ManiSkill3, it is performed every 500,000 steps.

The following describes the policy interaction process.

For MetaWorld, we first perform $n$ rounds of population policy interactions, during which individual policies are sampled using the Thompson sampler. If the same individual is sampled again, we will skip the interaction. Subsequently, we conduct policy interactions guided by the human-designed reward function. Each interaction corresponds to one episode. For ManiSkill3, we use 16 environments for parallel sampling by default. 4 rounds of policy interactions are performed as one policy interaction cycle, resulting in 64 environment steps (the default number of interaction steps before each training iteration in the SAC implementation).

After the interaction process, the collected data is annotated with multiple rewards by the reward population, and the corresponding variable information are recorded for subsequent relabeling and other operations.

For training settings, we follow the configurations of SAC. Specifically, we set the UTD ratio to 1 for MetaWorld, and to 0.5 for ManiSkill3.

For the process of generating reward functions using the LLM, it is essential to extract necessary information, e.g., the human-designed reward function, the input variables, and the success criteria. Subsequently, we use Prompts 1 and 2 as input to the LLM to generate the initial population, with the suggestions from Prompt 3 appended to Prompt 2. Subsequently, during each evolution process, information about the best reward function is provided in Prompt 4, and the LLM is instructed to construct new individuals through reflection. The following describes the prompt design for the LLM.

---

**Prompt 1: Role Definition and Task Description**

You are a reward engineer trying to write reward functions to solve reinforcement learning tasks as effective as possible. Your goal is to write a reward function for the environment that will help the agent learn the task described in text. You can introduce or remove some reward components for better learning. Here is the current reward function. your task is to refine and enhance it: {task_target}

## Prompt 2: Human-Designed Reward Function and User Format

The reward function is defined as
compute_reward function
{task_obs_code_string_1}
_gripper_caging_reward function
{task_obs_code_string_2}
Based on the return values of the functions mentioned above, some criteria are determined as follows:
{criteria_code_string}
These variables are crucial for constructing your reward function. Note that it only applies to the return values of the functions mentioned above. Therefore, you should not modify the calculation methods for these variables in the functions, as it may disrupt the conditions for evaluation. Instead, focus on leveraging this information to design more efficient reward guidance.
Here are the modification suggestions
{suggestion}.
Please strictly follow them to rewrite the "compute_reward" and "_gripper_caging_reward" above separately, using the keys from the list below as function inputs. However, do not introduce any keys that are not present in the list.
{input_dict_string}
Repeatedly verify that all input variables in the function definition exist in the list, ensuring no errors in naming or the introduction of new variables.

**Prompt 3: Suggestions**

The code output should be formatted as a python code string: "```python ... ```". The return variables must be consistent with those provided in the given functions. You should neither add nor remove variables, nor modify their names. I will specifically search for the "compute_reward" and "_gripper_caging_reward" reward functions.

Please carefully consider the sub-tasks that need to be completed sequentially to achieve the current task, and determine what rewards are necessary for guiding each task.

Carefully read the logic of the code above and improve the code in three ways, each of which must include the following:

(1) Reward Components: Add or remove certain components. If there are no modifications, please provide a brief reason. for example, add xxx reward component to encourage the agent to do xxx and apply a weight x for better xxx

(2) Reward Weights: Adjust the weight of certain reward components or change the reward coefficients. If there are no modifications, please provide a brief reason. for example, change the reaching reward weight from 5.0 to 10.0 for better xxx

(3) Reward Calculation: Modify the reward calculation methods. If there are no modifications, please provide a brief reason. for example, change the reaching or catching reward calculation method or add exp to xxx reward component to encourage the agent to do xxx

Finally, summarize three areas of improvement and provide valid reasons for their effectiveness.

Some helpful tips for writing the reward function code:

(1) You may find it helpful to normalize the reward to a fixed range by applying transformations like np.exp to the overall reward or its components.

(2) If you choose to transform a reward component, then you must also introduce a temperature parameter inside the transformation function; this parameter must be a named variable in the reward function and it must not be an input variable. Each transformed reward component should have its own temperature variable.

(3) Please do not simply transform the reward components or adjust the hyperparameters. Some unnecessary reward components can be removed, while some components that may be effective for learning can be added to the final reward.

(4) Make sure the type of each input variable is correctly specified; All the necessary information is provided in the function inputs, and "self" is neither referenced nor called.

(5) Do not modify the conditions for determining success, proximity to the object, or object grasping, as this would compromise the evaluation criteria.

(6) It is necessary to adjust some parameters of the existing reward function, such as scaling the reward for grasping, scaling the proximity reward, or scaling the success reward. For example "If condition: reward += 1.0 to reward +=0.5 for higher importance".

(7) If an error occurs while calling a function, implement the desired functionality based on your understanding instead of repeatedly calling the function.

**Prompt 4: Reflection Tips**

Based on the above reward function, the current RL policy's win rate is {win_rate}, the human designed expert reward is {current_score}, and the reward from the currently designed function is {current_our_score}. Below are the scores of the current policy on different metrics across multiple rounds during the evaluation process: {current_output}

Please carefully analyze the policy feedback. Some helpful tips for analyzing the policy feedback:

(1) If the success rates are always near zero, then you must rewrite the entire reward function.

(2) If the current policy has already performed well on certain metrics, the focus should shift to the subsequent tasks.

(3) If the reward is excessively large, it may need to be appropriately scaled to avoid learning issues.

# C  Generated Reward Examples by LaRes

Below is an example of the reward function constructed by LaRes, which successfully guided policy learning to achieve nearly 100% success rate on the Hammer task in its final stages. The following three contexts represent the main reward function, the subfunction and the improvement summaries. It can be observed that the LLM is capable of thoroughly analyzing the current issues and providing constructive improvement ideas.

**Generated Reward Function**

```python
def compute_reward(action, obs, HAMMER_HANDLE_LENGTH,
    _target_pos, NailSlideJoint_qpos, left_pad, right_pad,
    tcp_center, obj_init_pos, init_tcp):
    hand = obs[:3]
    hammer = obs[4:7]
    hammer_head = hammer + np.array([0.16, 0.06, 0.0])
    # Hammer position adjustment for proximity consideration
    hammer_threshed = hammer.copy()
    threshold = HAMMER_HANDLE_LENGTH / 2.0
    if abs(hammer[0] - hand[0]) < threshold:
        hammer_threshed[0] = hand[0]
    # Quaternion error reward calculation
    ideal = np.array([1.0, 0.0, 0.0, 0.0])
    error = np.linalg.norm(obs[7:11] - ideal)
    reward_quat = max(1.0 - error / 0.3, 0.0)  # Modified the
        denominator for stricter orientation
    # Computing caging and gripping rewards
    caging_and_gripping, caging = _gripper_caging_reward(
        left_pad,right_pad,tcp_center,obj_init_pos,
        init_tcp,action,hammer_threshed,
        object_reach_radius=0.01,obj_radius=0.015,
        pad_success_thresh=0.02,xz_thresh=0.01,high_density=True,
    )
    reward_grab = (caging_and_gripping + caging) / 2.0
    # Lifting reward with enhanced importance
    lifted = hammer_head[2] > 0.02
    pos_error = _target_pos - hammer_head
    a = 0.5   # Increased importance of lifting the hammer
    b = 1.5   # Increased importance of hitting the target nail
    reward_in_place = a * float(lifted) + b *
        reward_utils.tolerance(
        np.linalg.norm(pos_error),
        bounds=(0, 0.015),   # Reduced bounds for improved
            precision
        margin=0.1,
        sigmoid="long_tail")
    # Overall reward normalization
    temperature = 5.0   # Adjusted for better normalization
    reward = (2.0 * reward_grab + 4.0 * reward_in_place) *
        reward_quat / temperature
    # Success condition evaluation
    success = NailSlideJoint_qpos > 0.09
    if success and reward > 0.5:   # Reducing the threshold for
        checking success
        reward = 10.0
    return (reward, reward_grab, reward_quat, reward_in_place,
        success)
```

**Generated Subfunction**

```python
def _gripper_caging_reward(
        left_pad,
        right_pad,
        tcp_center,
        obj_init_pos,
        init_tcp,
        action,
        obj_pos,
        obj_radius=0.015,
        pad_success_thresh=0.02,
        object_reach_radius=0.01,
        xz_thresh=0.01,
        desired_gripper_effort=1.0,
        high_density=False,
        medium_density=False,
):
    """Reward for agent grasping the hammer."""
    if high_density and medium_density:
        raise ValueError("Can only be either high_density or
            medium_density")
    # Calculate distances for caging rewards
    pad_y_lr = np.hstack((left_pad[1], right_pad[1]))
    pad_to_obj_lr = np.abs(pad_y_lr - obj_pos[1])
    pad_to_objinit_lr = np.abs(pad_y_lr - obj_init_pos[1])
    # Define caging rewards based on y-axis
    caging_lr_margin = np.abs(pad_to_objinit_lr
        -pad_success_thresh)
    caging_lr =[reward_utils.tolerance(pad_to_obj_lr[i],bounds=
        (obj_radius, pad_success_thresh),
        margin=caging_lr_margin[i], sigmoid="long_tail",) for i
            in range(2)]
    caging_y = reward_utils.hamacher_product(*caging_lr)
    # Caging reward for x-z reference
    tcp, xz = tcp_center, [0, 2]
    caging_xz_margin = np.linalg.norm(obj_init_pos[xz] -
        init_tcp[xz]) - xz_thresh
    caging_xz = reward_utils.tolerance(np.linalg.norm(tcp[xz]-
        obj_pos[xz]),
        bounds=(0,xz_thresh),margin=caging_xz_margin,
        sigmoid="long_tail",)
    # Determine gripper closure extent
    gripper_closed = min(max(0, action[-1]),
        desired_gripper_effort)/desired_gripper_effort
    # Combining the various caging rewards
    caging = reward_utils.hamacher_product(caging_y, caging_xz)
    gripping = gripper_closed if caging > 0.97 else 0.0
    caging_and_gripping = reward_utils.hamacher_product(caging,
        gripping)
    if high_density:
        return caging_and_gripping, caging
    if medium_density:
        tcp = tcp_center
        tcp_to_obj = np.linalg.norm(obj_pos - tcp)
        tcp_to_obj_init = np.linalg.norm(obj_init_pos -
            init_tcp)
        reach_margin = abs(tcp_to_obj_init -
            object_reach_radius)
        reach = reward_utils.tolerance(tcp_to_obj,bounds=(0,
            object_reach_radius),margin=
            reach_margin,sigmoid="long_tail" )
        return caging_and_gripping, reach
```

**Improvement Summary**

### Summary of Improvements
1. **Reward Components**: The reward for lifting the hammer ('a') and for hitting the target nail ('b') was increased significantly to enhance the focus on these critical actions. The increased penalties when nearing the target position (e.g. adjusted bounds to '0.015') will encourage more precise interactions with the target area.
2. **Reward Weights**: The weight of the successful lift and placement were modified to provide enhanced emphasis on these actions, thereby creating a stronger incentive for the agent to accomplish these milestones, which seems crucial considering the policy's current low success rate.
3. **Reward Calculation**: The overall normalization and adjustment of reward components now includes changes in thresholds and the temperature parameter, which should provide more stability and control in training. Lowering the temperature value improves the agent's control over the learning process and minimizes excessive reward scaling.
These adjustments are proposed to directly address the demonstrated need for improvements given the agent's historical performance metrics, particularly with a near-zero success rate. The enhanced focus on the essential tasks and tighter constraints on the rewards should facilitate a more efficient learning trajectory towards successful manipulation tasks.

## D   Additional Experiments

Table 4: Performance under different population sizes.

| Pop size | 2 | 5 | 10 |
|---|---|---|---|
| soccer | 0.48 | **0.71** | 0.63 |
| pick-out-of-hole | 0.38 | **0.81** | 0.65 |
| hammer | 0.55 | 0.80 | **0.85** |

**Parameter analysis experiment on population size.** The results are shown in Table 4. We observe that a population size of 5 generally yields the best performance. A larger population may introduce potential out-of-distribution issues, while a smaller population can limit the algorithm's exploration capacity.

Table 5: Performance under different numbers of interaction steps.

| Interaction steps | 100,000 | 200,000 | 400,000 |
|---|---|---|---|
| soccer | 0.68 | **0.71** | 0.55 |
| pick-out-of-hole | **0.83** | 0.81 | 0.60 |
| hammer | 0.53 | **0.80** | 0.69 |

**Parameter analysis experiment on interaction steps.** The results are shown in Table 5. We observe that a frequency of 200k generally performs well. A smaller evolution frequency may lead to insufficient training of the lower-level policy, while a larger frequency can result in under-exploration of the reward function search space.

Table 6: Performance under different elite sizes.

| Elite size | 1 | 2 | 3 | 4 |
|---|---|---|---|---|
| soccer | 0.63 | **0.78** | 0.71 | 0.75 |
| pick-out-of-hole | 0.73 | 0.67 | **0.81** | 0.57 |
| hammer | 0.72 | 0.75 | **0.80** | 0.68 |

**Parameter analysis experiment on elite size.** The results are shown in Table 6. We find that 3 generally yields the best results. An elite size that is too small tends to increase the risk of falling into suboptimal solutions.

Table 7: Performance comparison using different LLM backbones.

| Backbone | basketball | soccer | pick-out-of-hole | hammer |
|---|---|---|---|---|
| 4o-mini | 0.87 | 0.71 | 0.81 | **0.80** |
| deepseek-v3 | **0.88** | 0.68 | 0.58 | 0.63 |
| qwen-plus | 0.63 | **0.88** | **0.98** | 0.50 |

**Comparative experiment using different LLMs as backbones.** We conducted evaluations with different LLMs as backbones, including Qwen-plus, GPT-4o-mini, and DeepSeek-V3. The results are shown in Table 7. We observe that although different models exhibit some variation in performance across tasks, LaRes consistently achieves a high success rate regardless of the underlying LLM framework.

**Parameter analysis experiment on the weight of constraint loss.** The results are shown in Table 8. LaRes does not tune this hyperparameter; it is set to the default value of 1.0 across all tasks. While further tuning may lead to improved performance, we find that 1.0 is generally sufficient to achieve strong results.

Table 8: Parameter analysis of the weight of constraint loss.

| Coefficient | 10.0 | 1.0 | 0.1 | 0.01 |
|---|---|---|---|---|
| pick-out-of-hole | 0.60 | 0.81 | **0.85** | 0.78 |
| soccer | 0.64 | 0.71 | **0.81** | 0.74 |

