# OpenReview forum: "LaRes: Evolutionary Reinforcement Learning with LLM-based Adaptive Reward Search"
_NeurIPS.cc/2025/Conference — NeurIPS 2025 poster_

### Official Review · Reviewer_c6GE · 2025-06-30

**Clarity:** 2
**Significance:** 3
**Originality:** 2
**Rating:** 4
**Confidence:** 4

**Summary:**

The paper introduces LaRes, a hybrid framework combining evolutionary algorithms (EAs) and reinforcement learning (RL) with large language models (LLMs) to optimize reward functions for policy learning. LaRes uses LLMs to generate and evolve reward function populations, a shared replay buffer for sample efficiency, and mechanisms like Thompson sampling and parameter constraints to stabilize training. Experiments on 16 robotic manipulation tasks show LaRes outperforms strong baselines (e.g., SAC, ERL variants) in sample efficiency and final performance, though some tasks exhibit marginal gains over EvoRainbow.

**Questions:**

- Elite Criterion: How is the "elite" reward function/policy formally defined (e.g., success rate threshold, moving average performance)? Does the definition vary across tasks or remain consistent?
- Ablation on Human-Designed Rewards: Have experiments been conducted where the human-designed reward is excluded from the population, and how does this affect performance? Comparison with Eureka and Roska on their benchmarks is suggested. This would validate the necessity of human guidance.
- Normalization vs. Scaling: Why use reward scaling with elite statistics instead of normalizing all rewards to the Normal distribution? Does the current approach introduce unnecessary complexity?
- Hyperparameter Sensitivity of Parameter Constraints: Has the weight of the parameter constraint loss (Equation 4) been ablated to assess its impact on stability and performance?

**Ethical Concerns:**

["NO or VERY MINOR ethics concerns only"]

**Final Justification:**

The authors have addressed my concerns.

**Limitations:**

yes.

**Quality:**

3

**Strengths And Weaknesses:**

**Strengths**
- Novel Reward Function Optimization: By leveraging LLMs to iteratively evolve reward functions, LaRes addresses a critical gap in prior ERL work that focused on policy parameter search. The framework demonstrates tangible improvements over human-designed rewards, highlighting the value of automated reward engineering.
- Sample Efficiency Mechanisms: The shared replay buffer with reward relabeling and Thompson sampling-based interaction prioritization effectively reuses historical data and focuses on high-performing policies. This design reduces sample waste and enhances learning stability, as validated by ablation studies.
- Comprehensive Evaluation: The paper conducts rigorous experiments across two benchmarks (MetaWorld, ManiSkill3) and compares against diverse baselines, including RL, ERL, and LLM-based reward search methods. The results consistently show LaRes’s superiority.

**Weaknesses**
- Elite Definition and Terminology Ambiguity: The paper uses "elite" and "non-elite" to denote high-performing vs. low-performing reward functions/policies, but the criteria for defining "elite" (e.g., success rate, cumulative reward) are not explicitly standardized. This ambiguity could confuse readers, particularly when discussing policy inheritance and parameter constraints.
- Dependence on Human-Designed Rewards: While LaRes starts from human-designed rewards and retains them in the population, the paper lacks experiments exploring fully LLM-generated rewards from scratch (without human input). Comparisons to Eureka under such conditions—where reward functions are initialized without human guidance—would strengthen the claim of LLM-driven optimization.
- LLM Generation Dynamics: The process of LLM-generated reward function updates (e.g., how many new functions are created per iteration, which ones are replaced) is not clearly defined. For example, the paper states the population size is 5, but it is unclear whether LLMs generate all 5 functions or replace a subset.
- Notation and Clarity in Formulation: The description of the Thompson sampling mechanism (e.g., "i-th action" referring to policy selection) is ambiguous, as it could be misinterpreted as action-space elements rather than policy indices. Clarifying this notation earlier in the text would improve readability.
- Continual Learning and Plasticity Concerns: The paper mentions using the best policy’s parameters for initialization but does not address potential plasticity loss (e.g., catastrophic forgetting), a known issue in continual learning [1]. Compared to methods like RoSKA [2], which explicitly handle this, the framework could be enhanced in its theoretical robustness.

[1] Maintaining plasticity in deep continual learning

[2] Efficient Language-instructed Skill Acquisition via Reward-Policy Co-Evolution

---

> ### Author Rebuttal · Authors · 2025-07-31
>
> We appreciate the reviewer's valuable review and constructive comments, and we would like you to know that your questions provide considerably helpful guidance to improve the quality of our paper.
>
> We will try our best to address each of the concerns and questions raised by the reviewer below:
> **1. [Re:  How is the "elite" reward function/policy formally defined (e.g., success rate threshold, moving average performance)? Does the definition vary across tasks or remain consistent?)]**
>
> The criterion for defining an "elite" is the success rate. We evaluate each policy over 20 episodes to assess the quality of the corresponding reward function. This definition is applied consistently across all manipulation tasks.
>
> **2. [Re: Dependence on Human-Designed Rewards: While LaRes starts from human-designed rewards and retains them in the population, the paper lacks experiments exploring fully LLM-generated rewards from scratch (without human input). Comparisons to Eureka under such conditions—where reward functions are initialized without human guidance—would strengthen the claim of LLM-driven optimization. （Comparison with Eureka and Roska on their benchmarks is suggested. This would validate the necessity of human guidance.）]**
>
> We thank the reviewer for valuable suggestions. LaRes can also be applied in settings where no human-designed reward functions are available.
>
> To address the reviewer’s concern, we evaluated LaRes on several tasks used in Text2Reward, where no expert rewards are provided. The results—measured in terms of success rate and sample cost—are shown below:
>
> |  | **window-close** | **window-open** | **drawer-open** | **button-press** | **door-close** |
> | --- | --- | --- | --- | --- | --- |
> | LLM zero shot | 51%   | 388796 | 6%    | 419084 | 8%    | 762346 | 65%  | 560428 | 15%  | 156590 |
> | Eureka | 50%   | 303786 | 55%  |  657098 | 15%   | 731876 | 61%  | 924163 | 98%  | 283206 |
> | LaRes | 100% | 164850 | 100% | 358403 | 100% | 164850 | 100% | 112800 | 100% | 65626 |
>
> We observe that our method still significantly outperforms other methods.
>
> **3. [Re: Comparison with Eureka and Roska on their benchmarks is suggested. This would validate the necessity of human guidance.]**
>
> We thank the reviewer for the valuable suggestion. LaRes primarily emphasizes sample efficiency, which requires development based on off-policy RL. However, modifying existing algorithms within IsaacGym to support efficient off-policy learning is quite challenging.
>
> To address the reviewer’s concern, we instead conducted experiments on OpenAI Humanoid and Ant tasks based on MUJOCO, which are commonly used in both Eureka and Roska. These experiments are performed without any human-designed rewards, and each is limited to 1 million environment steps.
> The results are shown below, we have currently completed experiments up to 500K steps, and we will update the results for 1M steps before the discussion deadline.
>
> | 500k | Eureka | ROSKA | LaRes |
> | --- | --- | --- | --- |
> | Ant | 1.49 | 1.57 | **3.61** |
> | Humanoid | 1.91 | 2.53 | **2.71** |
>
>
> We observe that LaRes significantly outperforms both Eureka and Roska on the Ant and Humanoid tasks.
>
> **4. [Re: LLM Generation Dynamics: The process of LLM-generated reward function updates (e.g., how many new functions are created per iteration, which ones are replaced) is not clearly defined. For example, the paper states the population size is 5, but it is unclear whether LLMs generate all 5 functions or replace a subset.]**
>
> In our experiments, we set the population size to 5 and maintained 3 elites. If the RL policy guided by the human-design reward function is selected as an elite, we generate 3 new reward functions; otherwise, 2 new reward functions are generated. The individuals to be replaced are selected based on their success rate, in ascending order.
>
> **5. [Re: Notation and Clarity in Formulation: The description of the Thompson sampling mechanism (e.g., "i-th action" referring to policy selection) is ambiguous, as it could be misinterpreted as action-space elements rather than policy indices. Clarifying this notation earlier in the text would improve readability.]**
>
>  We thank the reviewer for the valuable suggestion. We improve the presentation in the revised version.
>
> **6. [Re: Continual Learning and Plasticity Concerns: The paper mentions using the best policy’s parameters for initialization but does not address potential plasticity loss (e.g., catastrophic forgetting), a known issue in continual learning [1]. Compared to methods like RoSKA [2], which explicitly handle this, the framework could be enhanced in its theoretical robustness.]**
>
> We thank the reviewer for the valuable input. As shown in our previous responses, we provide comparative experiments; here, we would like to offer some clarification.
>
> ROSKA is based on PPO and primarily emphasizes policy reuse, whereas LaRes is built upon off-policy RL and focuses on sample reuse from a finer-grained perspective. Moreover, the notion of continual learning in LaRes emphasizes maintaining learning stability and consistency when the reward function changes, thereby reducing performance fluctuations. This differs from conventional continual learning, which typically focuses on network plasticity.
>
> We appreciate the reviewer’s suggestion and will include this discussion in the revised version.
>
> **7. [Re: Normalization vs. Scaling: Why use reward scaling with elite statistics instead of normalizing all rewards to the Normal distribution? Does the current approach introduce unnecessary complexity?]**
>
> In practice, the dataset is constructed using a learn-from-scratch approach. While using a normal distribution is a possible direction, it poses a challenge: during the first generation, the population must train for a long time to collect sufficient data to characterize the current reward distribution. This makes it difficult to directly model the initial reward functions using a normal distribution.
>
> If we leave the first-generation reward functions unchanged and only map subsequent rewards to a normal distribution, this would lead to a distribution shift problem.
>
> Therefore, we analyze the reward distribution of elite individuals from the previous generation, and re-map the outputs of newly generated reward functions to match this distribution.
>
> **8. [Re: Hyperparameter Sensitivity of Parameter Constraints: Has the weight of the parameter constraint loss (Equation 4) been ablated to assess its impact on stability and performance?]**
>
> We thank the reviewer for the helpful suggestion. LaRes does not tune this hyperparameter; it is set to the default value of 1.0 across all tasks. While further tuning may lead to improved performance, we find that 1.0 is generally sufficient to achieve strong results.
>
> |  | 10.0 | 1.0 | 0.1 | 0.01 |
> | --- | --- | --- | --- | --- |
> | pick-out-of-hole | 0.60 | 0.81 | 0.85 | 0.78 |
> | soccer | 0.64 | 0.71 | 0.81 | 0.74 |
>
> ---
>
> We hope our replies have addressed the concerns the reviewer posed and shown the improved quality of the paper. **We are always willing to answer any of the reviewer's concerns about our work and we are looking forward to more inspiring discussions.**

---

> > ### Comment · Reviewer_c6GE · 2025-08-05
> >
> > Thank you for your response; my question has been partially resolved. I hope you could update the clarified content from your response into the manuscript. Additionally, there are a few points in your reply that I’m still unclear about.
> >
> > First, in the response to Q2, could you explain what each of these values represents? Why are some presented as percentages and others as integers?
> >
> > Second, I’m quite surprised that your proposed method can achieve a 100% success rate without expert rewards. Assuming the experimental results are accurate, I would appreciate more details, such as the number of training iterations and a further analysis of this outcome.
> >
> > Regarding Q3, while the current results are already promising, you mentioned that updated results with 1M will be provided, and I look forward to seeing those. I also wonder how you obtained the values for Eureka and Roska in the table.
> >
> > For Q6, please directly address how your method enables the network to learn new rewards during the continuous evolution of the reward function while avoiding the issue of diminished network plasticity.
> >
> > Lastly, for Q7, why does using a normal distribution lead to the absence of statistical information in the first round? Aren’t these statistics just scale and offset values? If they cannot be obtained, how does the method in the paper acquire them?

---

> > > ### Author Response · Authors · 2025-08-05
> > > **We sincerely thank the reviewer for the constructive discussion and thoughtful responses! (2/2)**
> > >
> > > 5. **[Re: Lastly, for Q7, why does using a normal distribution lead to the absence of statistical information in the first round? Aren’t these statistics just scale and offset values? If they cannot be obtained, how does the method in the paper acquire them?]**
> > >
> > > To obtain the distribution of a reward function, a certain amount of interaction data is first required. Based on this data, the rewards can be computed, allowing us to estimate the mean and variance of the current reward function.
> > >
> > > However, before the first round of learning begins, the replay buffer is empty, and there is insufficient data to characterize the distribution of the reward function. Therefore, during the first round, no scale is applied to the reward function.
> > >
> > > By the second round, sufficient data has been collected. At this point, we can compute the mean and variance of the elite reward function from the first round. For the reward functions generated in the second round, we can calculate their respective means and variances based on the same data distribution. This allows us to determine the scale and offset needed to align each second-round reward function’s distribution with that of the elite reward function from the first round.
> > >
> > > This approach eliminates the need to predefine the reward distribution. Instead, in each iteration, we align new reward functions with the elite reward distribution from the previous generation using the same data distribution.
> > >
> > > ---
> > >
> > > **If the reviewer has different perspectives or alternative interpretations regarding our responses, we would be sincerely grateful for the opportunity to engage in further discussion. We also greatly appreciate any additional suggestions that could help improve this work.**
> > >
> > > **Thank you again for your thorough feedback and valuable time.**

---

> > > > ### Comment · Reviewer_c6GE · 2025-08-07
> > > >
> > > > After reading your response, I still have some lingering questions:
> > > >
> > > > Regarding the second question, I’m puzzled as to why the standard SAC was able to achieve a 100% success rate on this task *without* the provision of an expert reward. To clarify specifically, what I refer to as an "expert reward" here is the reward function inherent to the task itself. Methods like Eureka and Roska do not utilize this as their initial reward function, whereas the approach proposed by the authors does. My original question was intended to inquire about how the authors’ method would perform *without* this reward function. However, the response indicated that even the standard SAC could achieve 100% success without it, which leads me to worry that there might still be a misunderstanding regarding the experimental setup. Could the authors please elaborate further on this? It would be helpful to hear details about the specific experimental configurations, the inputs utilized by different methods, and an analysis of the experimental results. Additionally, could the authors explain why their implementation of SAC performed poorly? If SAC exhibits such significant reproducibility issues, I am concerned whether the authors’ method might also be susceptible to similar problems.
> > > >
> > > > With respect to the response to question 5, I would like to ask whether the approach adopted in the paper is able to obtain the mean and variance of the elite set right from the start of the first round. If so, could these not be used directly for normalization? I find it difficult to understand why Equation 3 in the paper is viable, whereas modifying it to a normal distribution—without introducing any new statistics—would pose issues.

---

> > > > > ### Author Response · Authors · 2025-08-07
> > > > > **Sincerely thank the reviewer for the constructive discussion (2/3)**
> > > > >
> > > > > We would like to assure the reviewer that there should be no concerns about the reproducibility of our results. Our implementation is robust, and **we are fully committed to releasing the code for transparency and reproducibility.**
> > > > >
> > > > > [1]. EvoRainbow: Combining Improvements in Evolutionary Reinforcement Learning for Policy Search. ICML 2024.
> > > > >
> > > > > [2]. Text2Reward: Reward Shaping with Language Models for Reinforcement Learning. ICLR 2024.
> > > > >
> > > > > 4. **[Re: With respect to the response to question 5, I would like to ask whether the approach adopted in the paper is able to obtain the mean and variance of the elite set right from the start of the first round. If so, could these not be used directly for normalization? I find it difficult to understand why Equation 3 in the paper is viable, whereas modifying it to a normal distribution—without introducing any new statistics—would pose issues.]**
> > > > >
> > > > > The approach adopted in the paper **can not obtain the mean and variance** of the elite set right from the start of the first round, primarily due to two reasons:
> > > > >
> > > > > - **Lack of interaction data**: Before the first round, there is a lack of interaction data, which means we can not characterize the reward distribution. As a result, it is not possible to compute the mean and variance.
> > > > > - **Unable to identify the elite:** Before any training occurs, the quality of reward functions cannot be evaluated. Therefore, it is not possible to determine which reward function should be considered elite.
> > > > >
> > > > > LaRes does not perform reward scaling in the first generation; it applies scaling only from the second generation onward. The purposes of reward scaling in LaRes include two main aspects:
> > > > >
> > > > > - Since the elite reward function has already demonstrated effectiveness under its corresponding distribution, aligning the distribution of newly generated reward functions with that of the elite one may be beneficial.
> > > > > - Since the replay buffer remains the same in LaRes, having a consistent reward distribution before and after the reward function update is more conducive to stable policy learning.
> > > > >
> > > > > Therefore, Equation (3) is only triggered when the elite reward function from the previous generation has been identified, and a new reward function needs to be distributionally aligned with it.
> > > > >
> > > > > - **Why is Equation (3) feasible?**
> > > > >
> > > > >     Thanks to the training in the previous generation, the replay buffer already contains sufficient interaction data. Based on this data, we can estimate the mean and variance of both the elite reward function and the newly generated reward function. Then, using Equation (3), we can align the distribution of the new reward function with that of the elite one.
> > > > >
> > > > > - **Why directly applying a normal distribution to all reward functions is problematic:**
> > > > >     1. As mentioned above, before the first generation, we are unable to obtain their mean and variance. These statistics can only be estimated starting from the second generation, based on the replay buffer accumulated during the previous generation. As a result, ****normal distribution can only be applied to later generations, which leads to a distribution shift between the first generation and the subsequent ones.
> > > > >     2. LaRes evaluates elite reward functions, which can reflect the quality of their associated reward distributions. In contrast, when using a normal distribution, it is unclear whether the resulting rewards are actually beneficial for learning. In fact, expert-designed reward functions often do not follow a normal distribution.
> > > > > - **Potential misunderstanding: Different between PPO and SAC**
> > > > >
> > > > >     There may be a misconception here that we would like to clarify: **LaRes prioritizes sample efficiency and is built on the off-policy SAC algorithm, rather than PPO.** PPO improves the policy based on advantages, which are inherently a relative measure. This makes it highly compatible with dynamically updating the mean and variance of rewards.
> > > > >
> > > > >     In contrast, SAC approximates the absolute value of the return through its value function. Dynamically modifying the reward during training can lead to instability or even collapse of the learning process.
> > > > >
> > > > >     Therefore, **dynamically applying a normal distribution to reward functions may be feasible for on-policy methods like PPO**—indeed, this is a commonly used technique in PPO [3]. However, **it is not applicable to off-policy algorithms such as SAC**.
> > > > >
> > > > >     [3]. The 37 Implementation Details of Proximal Policy Optimization

---

> > > > > ### Author Response · Authors · 2025-08-07
> > > > > **Sincerely thank the reviewer for the constructive discussion (3/3)**
> > > > >
> > > > > We have revisited the reviewer’s earlier question:
> > > > >
> > > > > **"Why does using a normal distribution lead to the absence of statistical information in the first round?"**
> > > > >
> > > > > We would like to clarify that it is not the use of a normal distribution that causes the absence of statistical information. Rather, it is the lack of sufficient data before the first generation that makes it impossible to accurately estimate the mean and variance.
> > > > >
> > > > > **We believe there might be some misunderstanding. We have tried our best to interpret the issue from various possible perspectives. If there is still anything unclear, we would be truly grateful for further clarification from the reviewer.**
> > > > >
> > > > > ---
> > > > >
> > > > > **Once again, we sincerely thank the reviewer for their response. We hope the above clarifications address your concerns. If we have misunderstood any part of your comments, we would greatly appreciate your further guidance. Thank you again for your valuable time and constructive discussion.**

---

> ### Author Response · Authors · 2025-08-05
> **We sincerely thank the reviewer for the constructive discussion and thoughtful responses! (1/2)**
>
> We are pleased to have addressed several of the concerns raised. Below, we provide our replies to the remaining issues.
>
> 1. **[Re: could you explain what each of these values represents? Why are some presented as percentages and others as integers?]**
>
> We sincerely apologize, we have just discovered a misalignment issue in the previously provided table. The correct results are as follows: the first value denotes the success rate, and the second value indicates the sample cost required to achieve that success rate. As shown, LaRes demonstrates superior sample efficiency.
>
> |  | **window-close** | **window-open** | **drawer-open** | **button-press** | **door-close** |
> | --- | --- | --- | --- | --- | --- |
> | LLM zero shot | 51%   \| 388796 | 6%    \| 419084 | 8%    \| 762346 | 65%  \| 560428 | 15%  \| 156590 |
> | Eureka | 50%   \| 303786 | 55%  \|  657098 | 15%   \| 731876 | 61%  \| 924163 | 98%  \| 283206 |
> | LaRes | 100% \| 164850 | 100% \| 358403 | 100% \| 164850 | 100% \| 112800 | 100% \| 65626 |
>
> 2. **[Re: Second, I’m quite surprised that your proposed method can achieve a 100% success rate without expert rewards. Assuming the experimental results are accurate, I would appreciate more details, such as the number of training iterations and a further analysis of this outcome.]**
>
> This results is expected, as these tasks are less challenging than those presented in the main body of our paper. Most of them are derived from the Text2Reward paper, where even a standard SAC algorithm can achieve a 100% success rate. In contrast, SAC performs poorly on the tasks proposed in our main experiments, often achieving very low or even zero success rates.
>
> The experimental settings are as follows: a population size of 5, a sample cost of 200,000 per iteration, and five iterations in total. All methods are based on the same SAC backbone, use the same prompt design with the same set of provided variables, ensuring that performance differences are not attributed to other confounding factors.
>
> 3. **[Re: while the current results are already promising, you mentioned that updated results with 1M will be provided, and I look forward to seeing those. I also wonder how you obtained the values for Eureka and Roska in the table.]**
>
> We sincerely appreciate the reviewers’ patience. Below, we present the results with 1M samples.
>
> | 1M | Eureka | ROSKA | LaRes |
> | --- | --- | --- | --- |
> | Ant | 1.49 | 2.77 | **4.44** |
> | Humanoid | 2.02 | 2.95 | **3.42** |
>
> Both Eureka and Roska are reproduced within the same framework as ours to ensure a fair comparison. As Roska has not released its source code, our implementation is based primarily on the configurations and pseudocode provided in its original paper and appendix. All algorithms are implemented on top of SAC. Overall, Eureka fails to ensure sustained performance improvement, whereas both ROSKA and LaRes consistently achieve such improvement, with LaRes exhibiting superior sample efficiency.
>
> 4. **[Re: For Q6, please directly address how your method enables the network to learn new rewards during the continuous evolution of the reward function while avoiding the issue of diminished network plasticity.]**
>
> We thank the reviewer for raising this question. We believe it is a valuable point. In our current work, we do not explicitly address the issue of plasticity. This is primarily based on two considerations:
>
> - As noted in paper [1], "Loss of plasticity is most relevant for systems that use small or no replay buffers, as large buffers can hide the effect of new data." Both Eureka and Roska are built on PPO, where the impact of plasticity loss tends to be more significant. In contrast, LaRes maintains a large replay buffer, which alleviates the severity of plasticity challenges [2].
> - When the reward function changes, LaRes relabels all experiences in replay buffer, which in turn changes the data distribution. Although we align reward distributions, data-level differences remain inevitable. Such changes in data can help mitigate plasticity degradation to some extent from a data-centric perspective. Some works have shown that modifying the replay buffer can help mitigate the loss of plasticity [3].
>
> We fully agree that explicitly addressing plasticity, as Roska does, is valuable and meaningful. However, it is beyond the current scope of our work. LaRes primarily focuses on improving sample efficiency at the sample level and reducing instability during the learning process.
>
> We will incorporate a discussion of plasticity-related works, particularly in connection with Roska, in the revised version. If the reviewer has any suggestions or recommendations, we would be grateful to receive them.
>
> [1]. Loss of plasticity in deep continual learning
>
> [2]. Experience Replay Addresses Loss of Plasticity in Continual Learning
>
> [3]. The Courage to Stop: Overcoming Sunk Cost Fallacy in Deep Reinforcement Learning

---

> ### Author Response · Authors · 2025-08-07
> **Sincerely thank the reviewer for the constructive discussion (1/3)**
>
> We greatly appreciate your thoughtful and detailed comments. Below, we provide clarifications regarding several of the concerns.
>
> 1. **[Re: I’m puzzled as to why the standard SAC was able to achieve a 100% success rate on this task without the provision of an expert reward.  My original question was intended to inquire about how the authors’ method would perform without this reward function. However, the response indicated that even the standard SAC could achieve 100% success without it]**
>
> There seems to be a misunderstanding here. **The claim that** **SAC can achieve a 100% success rate is based on the human-designed reward function, specifically the expert reward used in MetaWorld.**
>
> Below are the success rates and sample cost of SAC based on the human-designed reward in these tasks.
>
> |  | **window-close** | **window-open** | **drawer-open** | **button-press** | **door-close** |
> | --- | --- | --- | --- | --- | --- |
> | SAC w/ expert reward | 100%   \| 185500 | 100%    \| 164500 | 100%    \| 199500 | 100%  \| 164500 | 100%  \| 105,000 |
>
> These tasks are relatively simple, so SAC with expert-designed rewards can already achieve strong performance.
>
> We will further discuss the task difficulty levels in MetaWorld in our third response below.
>
> 2. **[Re: The inputs utilized by different methods, and an analysis of the experimental results]**
>
> The results below do not rely on human-designed rewards.
>
> | **w/o Expert Reward** | **window-close** | **window-open** | **drawer-open** | **button-press** | **door-close** |
> | --- | --- | --- | --- | --- | --- |
> | LLM zero shot | 51%   \| 388796 | 6%    \| 419084 | 8%    \| 762346 | 65%  \| 560428 | 15%  \| 156590 |
> | Eureka | 50%   \| 303786 | 55%  \|  657098 | 15%   \| 731876 | 61%  \| 924163 | 98%  \| 283206 |
> | LaRes | 100% \| 164850 | 100% \| 358403 | 100% \| 164850 | 100% \| 112800 | 100% \| 65626 |
>
> **All three methods, including LaRes, generate reward functions directly based on task descriptions, prompts, and available environment variables—without using any human-designed rewards. All three methods are based on exactly the same input for fair comparison.**
>
> LLM zero-shot refers to generating the reward function using the LLM only once, without any further tuning or feedback. Based on the results, we draw the following three conclusions:
>
> - Since the tasks belong to the simpler category, LLM zero-shot can typically achieve a certain level of success.
> - An evolutionary mechanism is essential—Eureka generally outperforms LLM zero-shot.
> - LaRes places greater emphasis on sample-level reuse, making it more sample-efficient than other methods—an aspect of primary concern in the DRL community.
> - LaRes w/o expert-designed rewards achieves even greater sample efficiency than SAC w/ expert-designed rewards in some tasks.
> 3. **[Re: Could the authors explain why their implementation of SAC performed poorly? If SAC exhibits such significant reproducibility issues, I am concerned whether the authors’ method might also be susceptible to similar problems.]**
>
> **Our SAC implementation directly adopts the open-source code provided in EvoRainbow [1], specifically the MetaWorld SAC implementation. The experimental results we obtained are consistent with those reported in the original paper, ensuring reproducibility.**
>
> **The SAC implementation used in our work demonstrates superior performance**. A comparison between our SAC results and those reported in Text2Reward reveals that **our SAC version consistently achieves better performance.**
>
> | **w/ expert reward** | **window-close** | **window-open** | **button-press** | **door-close** |
> | --- | --- | --- | --- | --- |
> | Our SAC  | 100%   \| 185500 | 100%    \| 164500 | 100%  \| 164500 | 100%  \| 105,000 |
> | SAC in Text2Reward | 100% \| 400k+ | about 42% \| 1M | 95% \| 600k+ | 100% \| about 300k |
>
> As for the tasks presented in the main text, the relatively poor performance of SAC with expert-designed rewards is primarily due to the high difficulty of these tasks.
>
> In MetaWorld, tasks can generally be categorized into three levels: **easy, medium, and hard**.
>
> - The tasks mentioned above—where reward functions are constructed directly without human-provided rewards—mostly fall into **the easy category**. In such tasks, SAC can achieve a 100% success rate using a human-designed reward function with very limited sample cost. Many reward generation works, such as Text2Reward[2], evaluate their methods under this category using zero-shot reward generation.
> - **The medium tasks** represent a more challenging group, where SAC with expert-designed rewards typically achieves less than 50% success rate after 1 million steps.
> - **The hard tasks** are the most difficult, where the success rate of SAC after 1 million steps is usually below 10%.
>
> **All the tasks included in the main text of our paper are selected from the medium and hard categories.**

---

> ### Comment · Reviewer_c6GE · 2025-08-08
>
> Thank you for your detailed response, which has clarified my doubts. Based on our discussion and the original manuscript, I believe this work demonstrates a certain degree of innovation, and the experimental results and conclusions hold some reference value. However, it appears that due to time constraints, the authors have not conducted supplementary experiments on tasks of medium or hard difficulty. Therefore, these experimental conclusions should be accompanied by rigorous descriptions of their prerequisite conditions. On the premise that the reviewers commit to incorporating the content discussed into the revised version, I will revise my rating upwards.

---

> > ### Author Response · Authors · 2025-08-08
> > **Sincere Thanks for the Reviewer’s In-Depth Discussion and Support!**
> >
> > We are very pleased to have addressed the reviewer’s concerns. We deeply appreciate the reviewer’s valuable time and recommendations, and sincerely thank you for your recognition and support of our work.
> >
> > The constructive suggestions are highly beneficial to improving the quality of our paper. In particular, we will incorporate the following aspects into the revised version:
> >
> > - Refine the manuscript’s presentation according to the reviewers’ comments, such as clarifying the definition of actions in Thompson sampling and specifying the setting of elite size.
> > - Incorporate all experiments discussed during the review, including the five MetaWorld tasks and the locomotion-related Ant and Humanoid tasks without human-designed rewards.
> > - Explicitly describe task difficulty levels and clearly state the prerequisite conditions for each set of experimental conclusions.
> > - Add a dedicated subsection discussing related work on continual learning and Roska.
> > - Include an ablation study analyzing the weight of the parameter constraint loss.
> > - Clarify the reward scaling mechanism and explain why direct normal-distribution normalization is not applied in the first generation.
> >
> > We commit that **all the above contents will be thoroughly and appropriately incorporated into the revised version**, and we also commit to **releasing the source code once the paper is made publicly available, with the repository link added to the manuscript**.
> >
> > Thanks to the reviewer's hard work and in-depth discussions with us.

---

### Official Review · Reviewer_XhCm · 2025-07-01

**Clarity:** 2
**Significance:** 3
**Originality:** 3
**Rating:** 4
**Confidence:** 3

**Summary:**

This paper proposes LaRes, an RL framework for LLM, which focuses on the reward function search. By applying reward function updates and Thompson selection mechanism, the proposed method can improve the historical data usage and prevent the policy collapse.

**Questions:**

1. Provide more experimental analysis on the computation cost.
2. Why LaRes exhibits larger variance compared to the baseline.

**Ethical Concerns:**

["NO or VERY MINOR ethics concerns only"]

**Final Justification:**

Overall, the proposed idea is interesting, but the efficiency on complex tasks has not been fully explored (although the authors provide some initial experiments in rebuttal). Also, the theoretical proof is not adequate. Thus, it is good for me to accept this work, but I will not champion it. I will keep my original score.

**Limitations:**

yes

**Quality:**

2

**Strengths And Weaknesses:**

Strength:

The proposed method applies LLM to achieve reward functions searching, which is an innovative framework.

Weakness

1. The relation between high-level reward evolution and low-level policy learning is not clear. Furthermore, the relation between the framework and techniques like continuous learning, Thompson sampling, and reward scaling should be clarified.

2. The authors claimed that LaRes can cooperate with other RL methods, which is not demonstrated in the evaluation. Moreover, the current design requires multiple Reward functions to fine-tune the model, some experiments on computation cost should be involved.

3. The main comparison in evaluation adopts SAC, which was proposed in 2018; more comparisons with SOTA methods are recommended. Furthermore, the baselines in ERL-ralated are out-of-date except ERL-Re and EvoRainbo.

4. In Fig2,3,4, several results of LaRes exhibit larger variance than baseline; more discussion on it is required.

5. No appendix, and the code is not open-sourced.

---

> ### Author Rebuttal · Authors · 2025-07-31
>
> We appreciate the reviewer's valuable review and constructive comments, and we would like you to know that your questions provide considerably helpful guidance to improve the quality of our paper.
>
> We will try our best to address each of the concerns and questions raised by the reviewer below:
> **1.[Re: The relation between high-level reward evolution and low-level policy learning is not clear. Furthermore, the relation between the framework and techniques like continuous learning, Thompson sampling, and reward scaling should be clarified.]**
>
> LaRes is primarily focused on sample-efficient task solving. To address this challenge, we decouple the problem into two levels: high-level reward function optimization and low-level policy learning. At the upper level, an LLM is used to construct a population of reward functions, which guide the learning of policies at the lower level. To further improve sample efficiency, we adopt a sample-level sharing strategy, where trajectories collected by different RL agents are shared and relabeled based on their respective reward functions.
>
> The introduction of Thompson Sampling also aims to enhance sample efficiency by allowing stronger policies within the RL population to interact more frequently with the environment, thereby reducing waste from weaker policies.
>
> It is important to note that our notion of continual learning differs from conventional definitions, which typically focus on network plasticity. In contrast, LaRes is designed to maintain policy stability in the face of evolving reward functions, ensuring that performance does not degrade or collapse when the reward function changes.
>
> To achieve this, we introduce reward scaling and a consistency loss. The former ensures distributional alignment of rewards before and after changes, while the latter constrains policy updates to prevent performance drops.
>
> **2. [Re: The authors claimed that LaRes can cooperate with other RL methods, which is not demonstrated in the evaluation. ]**
>
> LaRes is compatible with any off-policy RL algorithm. To address the reviewer’s concern, we integrated LaRes with DQN and evaluated it on MinAtar, a set of discrete control tasks based on image inputs. The experimental results are shown below:
>
> |  | **Breakout** | **Asterix** | **Freeway** | **Space_invaders** |
> | --- | --- | --- | --- | --- |
> | DQN | 13.52 | 21.86 | 2.96 | 11.55 | 38.89 | 53.93 | 23.18 | 65.38 |
> | LaRes | 21.87 | 31.28 | 14.13 | 28.18 | 50.87 | 57.24 | 42.27 | 84.72 |
>
> We observe that LaRes consistently leads to performance improvements.
>
> **3. [Re: the current design requires multiple Reward functions to fine-tune the model, some experiments on computation cost should be involved.]**
>
> LaRes requires optimizing multiple RL policies in parallel, each guided by a different reward function. Executing this process serially would indeed incur significant computational overhead. To address this, we implemented a distributed framework using a server–client architecture: sampling is performed on the server side, while each client independently optimizes its own policy. Overall, due to scheduling and communication overhead, LaRes incurs approximately 20% additional runtime.
>
> Regarding computational resources, all experiments on MetaWorld are conducted using CPU only. Each run consumes (pop size + 1) CPU cores—one for the server process and the remaining for individual worker processes corresponding to each policy in the population.
>
> **4. [Re: The main comparison in evaluation adopts SAC, which was proposed in 2018; more comparisons with SOTA methods are recommended. Furthermore, the baselines in ERL-ralated are out-of-date except ERL-Re and EvoRainbo. ]**
>
> We thank the reviewer for the insightful comment. LaRes is inherently flexible and can be built upon different underlying algorithms. In general, the stronger the backbone algorithm, the better the overall performance of LaRes.
>
> To address the reviewer’s concern, we conducted a comparison using two advanced RL algorithms: TD7 and CrossQ. The results are shown below.
>
> |  | Coffee-Pull | Pick-out-of-hole | Soccer | Hammer |
> | --- | --- | --- | --- | --- |
> | TD7 | 0.65 | 0.0 | 0.53 | 0.82 |
> | CrossQ | 0.60 | 0.0 | 0.59 | 0.23 |
> | LaRes | 0.82 | 0.81 | 0.71 | 0.80 |
>
> We observe that although LaRes uses the relatively weaker SAC as its backbone, it still outperforms the advanced RL algorithms TD7 and CrossQ. This highlights the critical role of reward function optimization in achieving strong performance.
>
> **5. [Re: In Fig2,3,4, several results of LaRes exhibit larger variance than baseline; more discussion on it is required. Why LaRes exhibits larger variance compared to the baseline]**
>
> We thank the reviewer for the helpful suggestion. The performance of LaRes is indeed influenced by the inherent randomness of the LLM. In some runs, suboptimal reward functions are generated, leading to reduced policy learning efficiency and increased variance in certain tasks. We believe this issue can be mitigated by either increasing the number of reward function optimization iterations or enlarging the reward function population, both of which allow for a more thorough search and make the framework more robust to the stochastic nature of the LLM.
>
> **6. [Re: No appendix, and the code is not open-sourced.]**
>
> We provide a detailed appendix, including Limitations and future work (Appendix A), Implementation details (Appendix B), Reward function examples (Appendix C)
>
> We commit to releasing the code as soon as the paper is made public, which we consider an important contribution of this work.
>
> **7. [Re: Further experiments on Locomotion tasks]**
>
> To further demonstrate the effectiveness of LaRes, we evaluate it on the two most challenging tasks in OpenAI MuJoCo: Ant and Humanoid. Notably, this version does not incorporate any human-designed reward function.
>
> We have currently completed experiments up to 500K steps, and we will update the results for 1M steps before the discussion deadline.
>
> | 500k | Eureka | ROSKA | LaRes |
> | --- | --- | --- | --- |
> | Ant | 1.49 | 1.57 | 3.61 |
> | Humanoid | 1.91 | 2.53 | 2.71 |
>
> We observe that LaRes also demonstrates superior performance on these tasks, further highlighting its effectiveness and generality.
>
> ---
>
> We hope our replies have addressed the concerns the reviewer posed and shown the improved quality of the paper. **We are always willing to answer any of the reviewer's concerns about our work and we are looking forward to more inspiring discussions.**

---

> > ### Author Response · Authors · 2025-08-05
> > **Update and Supplement of Experimental Results**
> >
> > We sincerely thank the reviewer for the valuable time. We would like to provide two updates regarding the experimental results, specifically focusing on the setting without human-designed reward functions:
> >
> > - Results on the MetaWorld benchmark (in each cell, the first value indicates the success rate, and the second denotes the corresponding sample cost).
> >
> > |  | **window-close** | **window-open** | **drawer-open** | **button-press** | **door-close** |
> > | --- | --- | --- | --- | --- | --- |
> > | LLM zero shot | 51%   \| 388796 | 6%    \| 419084 | 8%    \| 762346 | 65%  \| 560428 | 15%  \| 156590 |
> > | Eureka | 50%   \| 303786 | 55%  \|  657098 | 15%   \| 731876 | 61%  \| 924163 | 98%  \| 283206 |
> > | LaRes | 100% \| 164850 | 100% \| 358403 | 100% \| 164850 | 100% \| 112800 | 100% \| 65626 |
> > - Results on the Humanoid and Ant tasks (speed m/s).
> >
> > | 1M | Eureka | ROSKA | LaRes |
> > | --- | --- | --- | --- |
> > | Ant | 1.49 | 2.77 | **4.44** |
> > | Humanoid | 2.02 | 2.95 | **3.42** |
> >
> > Based on the above two experiments, we observe that LaRes demonstrates a clear advantage under settings without human-designed rewards. Furthermore, LaRes outperforms other baselines on higher-dimensional tasks such as Humanoid and Ant.
> >
> > **We sincerely appreciate the reviewer’s valuable comments and suggestions. If there are any remaining concerns, please do not hesitate to let us know. We will make every effort to address them.**

---

> ### Comment · Area_Chair_5c8W · 2025-08-06
> **Discuss rebuttal**
>
> Dear Reviewer XhCm,
>
> The author-reviewer discussion period will end soon on Aug. 8. Please read the authors' rebuttal and engage actively in discussion with the authors.
>
> AC

---

### Official Review · Reviewer_4fMG · 2025-07-03

**Clarity:** 3
**Significance:** 2
**Originality:** 3
**Rating:** 4
**Confidence:** 2

**Summary:**

This paper introduces LaRes, a novel framework that integrates Large Language Models (LLMs) with Evolutionary Reinforcement Learning (ERL) to iteratively search for and optimize reward functions instead of policy parameters—a departure from traditional ERL methods. The core idea is to evolve reward functions using LLMs and evaluate them via reinforcement learning performance, thus enhancing both sample efficiency and final task success.

**Questions:**

- Could the authors provide any theoretical insights into convergence properties or bounds for LaRes? How can we trust the stability of LLM-driven reward evolution in the long run?
- How does LaRes scale with more complex or high-dimensional tasks beyond the 16 robotic benchmarks tested?

**Ethical Concerns:**

["NO or VERY MINOR ethics concerns only"]

**Final Justification:**

The MinAtar, Ant, and Humanoid experimental results submitted by the authors convince me of LaRes’s potential in handling more complex or higher-dimensional tasks. Overall, I consider this an good paper and have therefore decided to maintain my score.

**Limitations:**

yes

**Quality:**

3

**Strengths And Weaknesses:**

Strengths:
- Shifting the focus from policy parameter search to reward function search using LLMs is both novel and impactful.
- The framework is presented clearly with diagrams (e.g., Figure 1) and step-by-step algorithmic breakdowns (Algorithm 1).

Weaknesses:
- The paper acknowledges that LLM-generated functions may lack diversity, which could limit the exploration potential of the evolutionary process.
- While the authors also explore scratch generation, the method generally relies on human-designed reward functions as a base, which may limit applicability in settings where expert knowledge is unavailable.

---

> ### Author Rebuttal · Authors · 2025-07-31
>
> We appreciate the reviewer's valuable review and constructive comments, and we would like you to know that your questions provide considerably helpful guidance to improve the quality of our paper.
>
> We will try our best to address each of the concerns and questions raised by the reviewer below:
> **1. [Re: The paper acknowledges that LLM-generated functions may lack diversity, which could limit the exploration potential of the evolutionary process]**
>
> Yes, we promote diversity by constructing an LLM-based population and employing a relatively high temperature during generation. While we acknowledge that more effective mechanisms for ensuring diversity may exist, this is not the primary focus of LaRes at the current stage.
>
> **2. [Re: While the authors also explore scratch generation, the method generally relies on human-designed reward functions as a base, which may limit applicability in settings where expert knowledge is unavailable.]**
>
> We thank the reviewer for the helpful suggestion. We conducted experiments on the following two domains where no human-designed reward function is required:
>
> - **5 Tasks from Metaworld (win rate & sample cost):**
>
> |  | **window-close** | **window-open** | **drawer-open** | **button-press** | **door-close** |
> | --- | --- | --- | --- | --- | --- |
> | LLM zero shot | 51%   | 388796 | 6%    | 419084 | 8%    | 762346 | 65%  | 560428 | 15%  | 156590 |
> | Eureka | 50%   | 303786 | 55%  |  657098 | 15%   | 731876 | 61%  | 924163 | 98%  | 283206 |
> | LaRes | 100% | 164850 | 100% | 358403 | 100% | 164850 | 100% | 112800 | 100% | 65626 |
> - **Openai Humanoid & Ant tasks:**
> We have currently completed experiments up to 500K steps, and we will update the results for 1M steps before the discussion deadline.
>
> | 500k | Eureka | ROSKA | LaRes |
> | --- | --- | --- | --- |
> | Ant | 1.49 | 1.57 | **3.61** |
> | Humanoid | 1.91 | 2.53 | **2.71** |
>
> We observe that LaRes still demonstrates a significant advantage.
>
> **3. [Re: Could the authors provide any theoretical insights into convergence properties or bounds for LaRes? How can we trust the stability of LLM-driven reward evolution in the long run?]**
>
> The results generated by LaRes are often superior to those produced by standard RL algorithms. This is primarily because LaRes follows the ERL framework and always maintains the expert reward function within the reward population. When LLM-generated reward functions perform poorly, the policy guided by the original reward function dominates the process, ensuring a performance lower bound. In addition, LaRes constructs an RL policy population, which further augments sample diversity and enhances overall performance.
>
> Our work primarily focuses on empirical validation of the proposed method's effectiveness, supported by extensive experiments across 12 MetaWorld tasks, 4 ManiSkill tasks, 5 human-reward-defined tasks, as well as Humanoid and Ant tasks. Regarding theoretical support, we acknowledge in the Limitations section that the current work lacks formal theoretical guarantees, which we consider an important direction for future research.
>
> **4. [Re: How does LaRes scale with more complex or high-dimensional tasks beyond the 16 robotic benchmarks tested?]**
>
> To address the reviewer’s concern, we primarily extended our experiments in two directions:
>
> - **MinAtar**, a benchmark based on image inputs.
> - **Ant & Humanoid**, two of the most challenging control tasks from OpenAI's suite.
>
> The results on **MinAtar**:
>
> |  | **Breakout** | **Asterix** | **Freeway** | **Space_invaders** |
> | --- | --- | --- | --- | --- |
> | DQN | 13.52 | 21.86 | 2.96 | 11.55 | 38.89 | 53.93 | 23.18 | 65.38 |
> | LaRes | 21.87 | 31.28 | 14.13 | 28.18 | 50.87 | 57.24 | 42.27 | 84.72 |
>
> The results on **Humanoid** and **Ant** without human-designed reward functions, we have currently completed experiments up to 500K steps, and we will update the results for 1M steps before the discussion deadline.
>
> | 500k | Eureka | ROSKA | LaRes |
> | --- | --- | --- | --- |
> | Ant | 1.49 | 1.57 | 3.61 |
> | Humanoid | 1.91 | 2.53 | 2.71 |
>
> We observe that LaRes also demonstrates superior performance on these tasks, further highlighting its effectiveness and generality.
>
> ---
>
> We hope our replies have addressed the concerns the reviewer posed and shown the improved quality of the paper. **We are always willing to answer any of the reviewer's concerns about our work** and we are looking forward to more inspiring discussions.

---

> > ### Author Response · Authors · 2025-08-05
> > **Update and Supplement of Experimental Results**
> >
> > Once again, we sincerely thank the reviewer for their valuable time. Here, we would like to provide updates on the following two points:
> >
> > - We recently discovered a formatting error in the previously submitted table (Experiments without human-deigned reward function). The corrected results are shown below.
> >
> > |  | **window-close** | **window-open** | **drawer-open** | **button-press** | **door-close** |
> > | --- | --- | --- | --- | --- | --- |
> > | LLM zero shot | 51%   \| 388796 | 6%    \| 419084 | 8%    \| 762346 | 65%  \| 560428 | 15%  \| 156590 |
> > | Eureka | 50%   \| 303786 | 55%  \|  657098 | 15%   \| 731876 | 61%  \| 924163 | 98%  \| 283206 |
> > | LaRes | 100% \| 164850 | 100% \| 358403 | 100% \| 164850 | 100% \| 112800 | 100% \| 65626 |
> >
> > - In addition, we have completed the 1M-step experiments on the Humanoid and Ant tasks. The results are as follows:
> >
> > | 1M | Eureka | ROSKA | LaRes |
> > | --- | --- | --- | --- |
> > | Ant | 1.49 | 2.77 | **4.44** |
> > | Humanoid | 2.02 | 2.95 | **3.42** |
> >
> > It can be seen that LaRes achieves better performance than other methods without relying on human-designed rewards.
> >
> > **We sincerely appreciate the reviewer’s valuable comments and suggestions. If there are any remaining concerns, please do not hesitate to let us know. We will make every effort to address them.**

---

> ### Comment · Area_Chair_5c8W · 2025-08-06
> **Discuss rebuttal**
>
> Dear Reviewer 4fMG,
>
> The author-reviewer discussion period will end soon on Aug. 8. Please read the authors' rebuttal and engage actively in discussion with the authors.
>
> AC

---

> ### Author Response · Authors · 2025-08-08
> **Sincere Thanks and Request for Reviewer’s Feedback**
>
> Dear Reviewer 4fMG
>
> Thank you very much for your valuable time and constructive feedback.
>
> As there is **only one day remaining before the author–reviewer discussion closes,** we would like to kindly confirm whether our responses have addressed your concerns, or if you have any further suggestions or remaining questions. Your additional comments will be greatly appreciated and will help us ensure the manuscript meets the highest possible standard.
>
> **We sincerely appreciate your input and look forward to your reply.**
>
> Best regards,
>
> Authors Paper 27186

---

### Official Review · Reviewer_bxvD · 2025-07-05

**Clarity:** 4
**Significance:** 3
**Originality:** 3
**Rating:** 4
**Confidence:** 3

**Summary:**

This paper proposes LaRes, a framework that uses LLMs within an evolutionary algorithm to search for optimal reward functions for RL tasks. By evolving reward functions directly, it aims to automate a critical part of the reinforcement learning workflow. It introduces several techniques like reward relabeling and parameter constraints to improve sample efficiency and stability.

**Questions:**

In addition to the points raised in the weakness section, I have three further academic questions for the authors:
1. In the framework, the elite reward functions are not replaced, and the RL agent guided by the reward function designed by humans does not participate in Thompson sampling to ensure the stability of learning. However, does this design also bring the risk of prematurely converging to local optima? If these strategies are not globally optimal, will the mechanism continuously guide the search process, thereby hindering the exploration of new and potentially better reward paradigms? Have you considered other mechanisms to balance stability and the need for long-term exploration?
2. The paper demonstrates that the final evolved reward functions are effective. However, what is the nature of this evolutionary trajectory? Do the reward functions converge towards a specific structure? Do they tend to become more complex over generations, or do they become simpler as the LLM distills the problem's essence?
3. Appendix A mentions that the current method relies on the randomness of LLMS to ensure diversity, but does not explicitly consider individual differences. Have you considered adding explicit instructions in the Prompt to guide the LLM to generate more diverse reward functions? Will this further enhance the performance of LaRes?

**Ethical Concerns:**

["NO or VERY MINOR ethics concerns only"]

**Final Justification:**

Thank you to the authors for the detailed rebuttal. I currently have no further questions. The additional experiments on different LLMs and hyperparameter sensitivity have resolved most of my technical concerns.

My overall suggestion is to better clarify the scope of the paper's contributions. The paper demonstrates that LaRes is highly effective on a wide range of complex manipulation tasks. However, given the authors' insightful discussion on the challenges of reward hacking and LLM bias, it is difficult to fully believe that the same method can readily solve hard-exploration problems where the necessary reward signals are non-intuitive.

**Limitations:**

See the above weakness section.

**Quality:**

3

**Strengths And Weaknesses:**

## Strengths:
It's always good to see novel approaches to the reward design problem in RL. I particularly appreciate the systematic framework design, which demonstrates sophisticated LLM integration through closed-loop optimization with quantitative feedback. The core idea of using an LLM as an evolutionary operator to search the reward function space is creative and could inspire follow-up work along this direction.
## Weaknesses:
There are a few points that I would like to suggest here to make the paper even stronger.
* Although paper shows that LLM-generated rewards improve performance significantly, hard-exploration tasks are often difficult because they require non-intuitive intermediate rewards. It would be insightful if the authors can discuss the relationships between the LLM's generative priors and performance on such tasks. Specifically, why might an LLM-guided search still fail on a hard-exploration task even if it can generate dense rewards? Does the LLM's common sense reasoning create a bias that prevents it from discovering the unconventional reward components needed to solve these problems?
* When the reward function population is updated, all experiences in the shared replay pool need to be relabeled. How much computational overhead is this process for a replay pool of samples of different sizes? I hope authors can conduct supplementary experiments for analysis and examine the trade-offs related to performance.
* LaRes contains a series of hyperparameters of the genetic algorithm, such as the population size of the reward function, the number of evolutions, the elite size, etc. However, it seems that the article does not discuss the sensitivity of the framework's performance to these hyperparameters. It is hoped that supplementary experiments can be conducted to illustrate this.
* This paper relies exclusively on GPT-4o mini as the core LLM engine, which raises questions about the generalizability of the findings to other model families, particularly mainstream open-weight models like Llama or Qwen.  Without validation across different LLM architectures, the broader applicability of the LaRes framework remains uncertain. I would strongly recommend that the authors conduct additional experiments with alternative LLM to demonstrate the robustness and generality of their approach.

---

> ### Author Rebuttal · Authors · 2025-07-31
>
> We appreciate the reviewer's valuable review and constructive comments, and we would like you to know that your questions provide considerably helpful guidance to improve the quality of our paper.
>
> We will try our best to address each of the concerns and questions raised by the reviewer below:
>
> **1. [Re: It would be insightful if the authors can discuss the relationships between the LLM's generative priors and performance on such tasks. Specifically, why might an LLM-guided search still fail on a hard-exploration task even if it can generate dense rewards? Does the LLM's common sense reasoning create a bias that prevents it from discovering the unconventional reward components needed to solve these problems?]**
>
> First, task decomposition is essential for a given task, as it can significantly reduce the difficulty of reward generation. Directly using LLMs to generate long and complex reward functions is often impractical.
>
> We fully agree with the reviewer’s observation that even with dense rewards, learning a successful policy may still fail. This is mainly due to a potential misalignment problem—dense rewards are not necessarily positively correlated with task success, leading to reward hacking. This issue arises from multiple factors. On one hand, individual reward components may be poorly designed—for example, critical reward terms may be underweighted and dominated by less important ones, or inappropriate activation functions may be used.
>
> Moreover, reward design can be viewed as a black-box optimization problem, where some reward components that seem intuitively useful may not be necessary for successful policy learning. This highlights the need for RL–driven evaluation of reward functions. It also implies that when a reward function fails to guide learning effectively, the capability of the RL algorithm itself is a non-negligible factor. For instance, the performance of SAC and AC can differ significantly under the same reward design.
>
> Therefore, we argue that reward functions generated by LLMs, like those manually designed by human experts, are inevitably prone to being suboptimal or even ineffective. Thus, iterative refinement and reflective optimization are often necessary to achieve reliable and aligned reward signals.
>
> **2. [Re: When the reward function population is updated, all experiences in the shared replay pool need to be relabeled. How much computational overhead is this process for a replay pool of samples of different sizes?.]**
>
> The computational overhead of this process is minimal. We directly relabel the replay buffer through code, without involving any gradient operations, resulting in very low complexity.
>
> We measure the relabeling time and find that relabeling 200,000 samples takes only 5.89 seconds, and the cost scales linearly with the number of samples.
>
> **3.[Re: LaRes contains a series of hyperparameters of the genetic algorithm, such as the population size of the reward function, the number of evolutions, the elite size, etc. ]**
>
> We thank the reviewer for the valuable feedback. Regarding the population size, we followed the common settings used in prior ERL works, setting it to 5, with 5 evolution number( same with Eureka). The elite size is set to 3.
>
> To address the reviewer’s concern, we conducted the following additional hyperparameter experiments:
>
> | Pop size | 2 | 5 | 10 |
> | --- | --- | --- | --- |
> | soccer  | 0.48 | **0.71** | 0.63 |
> | pick-out-of-hole | 0.38 | **0.81** | 0.65 |
> | hammer | 0.55 | 0.80 | **0.85** |
>
> We observe that a population size of 5 generally yields the best performance. A larger population may introduce potential out-of-distribution issues, while a smaller population can limit the algorithm’s exploration capacity.
>
> | **interaction steps** | 100.000 | 200.000 | 400.000 |
> | --- | --- | --- | --- |
> | soccer  | 0.68 | **0.71** | 0.55 |
> | pick-out-of-hole | **0.83** | 0.81 | 0.60 |
> | Hammer | 0.53 | **0.80** | 0.69 |
>
> For the evolution number, we define it in the number of environment interaction steps between two evolutionary updates.
>
> We observe that a frequency of 200k generally performs well. A smaller evolution frequency may lead to insufficient training of the lower-level policy, while a larger frequency can result in under-exploration of the reward function search space.
>
> | Elite size | 1 | 2 | 3 | 4 |
> | --- | --- | --- | --- | --- |
> | soccer | 0.63 | **0.78** | 0.71 | 0.75 |
> | pick-out-of-hole | 0.73 | 0.67 | **0.81** | 0.57 |
> | hammer | 0.72 | 0.75 | **0.80** | 0.68 |
>
> For the elite size, we found that 3 generally yields the best results. An elite size that is too small tends to increase the risk of falling into suboptimal solutions.
>
> **4. [Re: I would strongly recommend that the authors conduct additional experiments with alternative LLM to demonstrate the robustness and generality of their approach.]**
>
> We thank the reviewer for the valuable suggestion. We conducted evaluations across a diverse set of benchmarks, including Qwen-plus, GPT-4o-mini, and DeepSeek-V3. The experimental results are shown below.
>
> |  | basketball | soccer | pick-out-of-hole | hammer |
> | --- | --- | --- | --- | --- |
> | 4o-mini | 0.87 | 0.71 | 0.81 | 0.80 |
> | deepseek-v3 | 0.88 | 0.68 | 0.58 | 0.63 |
> | qwen-plus | 0.63 | 0.88 | 0.98 | 0.50 |
>
> We observe that although different models exhibit some variation in performance across tasks, LaRes consistently achieves a high success rate regardless of the underlying LLM framework.
>
> **5. [Re: In the framework, the elite reward functions are not replaced, and the RL agent guided by the reward function designed by humans does not participate in Thompson sampling to ensure the stability of learning. However, does this design also bring the risk of prematurely converging to local optima? If these strategies are not globally optimal, will the mechanism continuously guide the search process, thereby hindering the exploration of new and potentially better reward paradigms? Have you considered other mechanisms to balance stability and the need for long-term exploration?]**
>
> We agree with the reviewer's perspective that exploration versus exploitation is a fundamental challenge in any learning or optimization problem. We did consider using UCB to address this issue more effectively; however, UCB introduces several hyperparameters that require careful tuning. While better performance might be achievable through fine-tuning, we chose to adopt Thompson Sampling for the sake of simplicity.
>
> As for long-term exploration, one viable approach is to increase the number of environment interaction steps per generation for each population member. However, this presents a trade-off: too many steps may lead to sample inefficiency, while too few may result in suboptimal performance.
>
> Finally, regarding the decision to exclude the RL reward function from evolutionary optimization, this follows prior ERL settings. This design ensures that when the population fails to discover better solutions, the RL reward function can still provide a performance lower bound.
>
> **6. [Re: what is the nature of this evolutionary trajectory? Do the reward functions converge towards a specific structure? Do they tend to become more complex over generations, or do they become simpler as the LLM distills the problem's essence?]**
>
> In practice, we observe that the structure of the reward function typically remains relatively stable from the early to later stages of training. For example, in a pick-and-place task, the reward function usually consists of components such as approach reward, grasp reward, and reward for moving toward the target. The main changes over time tend to lie in the tuning of hyperparameters—both within and between reward components—as well as in the implementation details of each module, such as different transformation functions. Overall, the reward function does not become increasingly complex; instead, once the general structure is established, subsequent efforts focus on refining and optimizing specific details.
>
> **7. [Re: Have you considered adding explicit instructions in the Prompt to guide the LLM to generate more diverse reward functions? Will this further enhance the performance of LaRes? ]**
>
> We thank the reviewer for the valuable suggestion. In our implementation, we set a relatively high temperature for the LLM (i.e., 1.0) to promote population diversity.
>
> To address the reviewer’s concern, we experiment with designing diversity-enhancing prompts. Specifically, in each generation step, we feed the previously generated result back into the LLM as an additional input, encouraging the model to produce more diverse outputs. The results are shown below.
>
> |  | hammer | soccer | **pick-out-of-hole** |
> | --- | --- | --- | --- |
> | Diversity version | 0.60 | 0.68 | 0.60 |
> | Org version | 0.80 | 0.71 | 0.81 |
>
> We observe that this approach leads to a decline in performance. Upon closer inspection of the generated reward functions, we identify a key issue: while the method increases diversity, it often results in reward functions that omit essential modules. For example, after the first reward function is generated with a human-aligned structure, subsequent ones tend to introduce entirely different reward components for the sake of diversity.
>
> As previously discussed, good reward functions tend to share a common structure, differing mainly in their finer details. We therefore believe that enhancing diversity requires more principled mechanism design. Simply relying on prompt-based guidance is insufficient to ensure both high diversity and strong performance.
>
> ---
> We hope our replies have addressed the concerns the reviewer posed and shown the improved quality of the paper. **We are always willing to answer any of the reviewer's concerns about our work and we are looking forward to more inspiring discussions.**

---

> > ### Author Response · Authors · 2025-08-05
> > **Update and Supplement of Experimental Results**
> >
> > We sincerely thank the reviewer for the valuable time. We would like to provide two updates regarding the experimental results, specifically focusing on the setting without human-designed reward functions:
> >
> > - Results on the MetaWorld benchmark (in each cell, the first value indicates the success rate, and the second denotes the corresponding sample cost).
> >
> > |  | **window-close** | **window-open** | **drawer-open** | **button-press** | **door-close** |
> > | --- | --- | --- | --- | --- | --- |
> > | LLM zero shot | 51%   \| 388796 | 6%    \| 419084 | 8%    \| 762346 | 65%  \| 560428 | 15%  \| 156590 |
> > | Eureka | 50%   \| 303786 | 55%  \|  657098 | 15%   \| 731876 | 61%  \| 924163 | 98%  \| 283206 |
> > | LaRes | 100% \| 164850 | 100% \| 358403 | 100% \| 164850 | 100% \| 112800 | 100% \| 65626 |
> > - Results on the Humanoid and Ant tasks (speed m/s).
> >
> > | 1M | Eureka | ROSKA | LaRes |
> > | --- | --- | --- | --- |
> > | Ant | 1.49 | 2.77 | **4.44** |
> > | Humanoid | 2.02 | 2.95 | **3.42** |
> >
> > Based on the above two experiments, we observe that *LaRes* demonstrates a clear advantage under settings without human-designed rewards. Furthermore, it significantly outperforms other baselines on higher-dimensional tasks such as *Humanoid* and *Ant*.
> >
> > **We sincerely appreciate the reviewer’s valuable comments and suggestions. If there are any remaining concerns, please do not hesitate to let us know. We will make every effort to address them.**

---

> ### Comment · Area_Chair_5c8W · 2025-08-06
> **Discuss rebuttal**
>
> Dear Reviewer bxvD,
>
> The author-reviewer discussion period will end soon on Aug. 8. Please read the authors' rebuttal and engage actively in discussion with the authors.
>
> AC

---

### Comment · Area_Chair_5c8W · 2025-08-05
**Mandatory rebuttal acknowledgement**

Dear Reviewers 4fMG, XhXm, bvxD:

Please note “Mandatory Acknowledgement” button is to be submitted only when you fulfill all conditions below (conditions in the acknowledgment form):
* read the author rebuttal
* engage in discussions (reviewers must talk to authors, and optionally to other reviewers and AC - ask questions, listen to answers, and respond to authors)
* fill in "Final Justification" text box and update “Rating” accordingly (this can be done upon convergence - reviewer must communicate with authors first)

AC

---

> ### Comment · Area_Chair_5c8W · 2025-08-07
> **Response to authors' rebuttal and comments**
>
> Dear Reviewers,
>
> Please note “Mandatory Acknowledgement” button is to be submitted only when you fulfill all conditions below (conditions in the acknowledgment form):
> * read the author rebuttal
> * engage in discussions (reviewers must talk to authors, and optionally to other reviewers and AC - ask questions, listen to answers, and respond to authors)
> * fill in "Final Justification" text box and update “Rating” accordingly (this can be done upon convergence - reviewer must communicate with authors first)
>
> AC

---

### Decision · Program_Chairs · 2025-09-17

**Decision:**

Accept (poster)

**Comment:**

(a) Summary

This paper investigates how to improve reinforcement learning(RL) via evolutionary algorithms(EA) in the reward function space. It proposes to generate reward function population using LLM and EA for selecting the optimal reward to guide policy learning. It also introduces reward relabeling and parameter constraints to improve sample efficiency and stability. Experiments validate the effectiveness of the proposed method.

(b) Strengths
- Novel reward function optimization: The core idea of using an LLM as an evolutionary operator to search the reward function space is creative and could inspire follow-up work along this direction.
- Sample Efficiency Mechanisms: The shared replay buffer with reward relabeling and Thompson sampling-based interaction prioritization effectively reuses historical data and focuses on high-performing policies. This design reduces sample waste and enhances learning stability, as validated by ablation studies.
- Comprehensive Evaluation: The paper conducts rigorous experiments across two benchmarks (MetaWorld, ManiSkill3) and compares against diverse baselines, including RL, ERL, and LLM-based reward search methods.

(c) Weaknesses
- LLM-generated functions may lack diversity, which could limit the exploration potential of the evolutionary process.
- Dependence on Human-Designed Rewards: the method generally relies on human-designed reward functions as a base, which may limit applicability in settings where expert knowledge is unavailable.
- Computation cost: when the reward function population is updated, all experiences in the shared replay pool need to be relabeled.

(d) rebuttal

This is a borderline paper. The reviewers find the proposed approach novel and sample-efficient. The authors addressed the concerns on human-designed rewards, llm-generated functions, and computational overhead with additional experiments and clarification. Most reviewers remain positive after the rebuttal.

(e) Decision

Shifting the focus from policy parameter search to reward function search using LLMs is both novel and impactful; it could inspire follow-up work along this direction. The shared replay buffer reduces sample waste and enhances learning stability. I recommend acceptance.